# Aquila enables reference-assisted diploid personal genome assembly and comprehensive variant detection based on linked reads

Xin Zhou [1,4✉], Lu Zhang[2,5], Ziming Weng[2], David L. Dill[1] & Arend Sidow [2,3✉]

We introduce Aquila, a new approach to variant discovery in personal genomes, which is critical for uncovering the genetic contributions to health and disease. Aquila uses a reference sequence and linked-read data to generate a high quality diploid genome assembly, from which it then comprehensively detects and phases personal genetic variation. The contigs of the assemblies from our libraries cover >95% of the human reference genome, with over 98% of that in a diploid state. Thus, the assemblies support detection and accurate genotyping of the most prevalent types of human genetic variation, including single nucleotide polymorphisms (SNPs), small insertions and deletions (small indels), and structural variants (SVs), in all but the most difficult regions. All heterozygous variants are phased in blocks that can approach arm-level length. The final output of Aquila is a diploid and phased personal genome sequence, and a phased Variant Call Format (VCF) file that also contains homozygous and a few unphased heterozygous variants. Aquila represents a cost-effective approach that can be applied to cohorts for variation discovery or association studies, or to single individuals with rare phenotypes that could be caused by SVs or compound heterozygosity.

[1] Department of Computer Science, Stanford University, Stanford, CA, USA. [2] Department of Pathology, Stanford University, Stanford, CA, USA. [3] Department of Genetics, Stanford University, Stanford, CA, USA. [4] Present address: Department of Biomedical Engineering, Vanderbilt University, Nashville, TN, USA. [5] Present address: Department of Computer Science, Hong Kong Baptist University, Kowloon Tong, Hong Kong. ✉email: maizie.zhou@vanderbilt.edu; arend@stanford.edu

Despite recent advances, quantifying the contribution of genetic variation to specific disease risk is a stubborn biomedical problem that remains far from solved. In general, understanding the relationship between genotype and phenotype requires complete ascertainment of genotype, which for humans has yet to be achieved in a scalable fashion. At this stage in technology development, DNA sequencing still faces a vexing tradeoff between cost and completeness so that discovery of variation in larger cohorts is limited to SNPs and small indels. In fact, the relatively low cost of Illumina-based short-fragment whole genome sequencing and the even lower cost of exomes and genotyping arrays has caused considerable ascertainment bias such that the vast majority of genotype–phenotype associations focus on SNPs with small effect, even though the undetected larger variation is known to involve roughly as many bases in our genomes as SNPs and is therefore predicted to have significant phenotypic impact as well[1,2]. Also generally missing is the phasing of genetic variation, which is similarly important for estimation of phenotypic impact, as the distinction between *cis* and *trans* compound heterozygotes in an essential locus can mean the difference between health and disease[3] and is likely to modulate risk of multigenic disease as well[4].

Single-molecule sequencing approaches, particularly Pacific Biosciences (PacBio) and Oxford Nanopore Technologies (ONT), provide potential solutions, as long-range information allows accurate detection of SVs and phasing[5–7]. Despite recent improvements in base calling, the drawback of ONT is that it still exhibits lower base-pair level accuracy than Illumina. A widely applied solution has been to supplement long reads with higher quality short read data, but these ensemble approaches are difficult to scale to larger cohorts due to the complexity of data generation, integration, and analysis, and have therefore been limited to small sample sizes in proof-of-principle studies[8,9]. A solution to making long reads more accurate is to sequence the same single molecule multiple times to reduce error, as implemented in the PacBio circular consensus sequencing approach, now called HiFi[10–12]. However, HiFi requires large amounts of input DNA and several-fold oversampling of the same molecule, a currently expensive proposition for anything but small sample sizes.

A relatively recent addition to the DNA-sequencing ecosystem has been pioneered by 10X Genomics, wherein the original large molecules of a gentle DNA preparation are partitioned into microfluidic compartments[13,14]. Via a series of within-compartment molecular biology and subsequent standard steps of library construction and sequencing, barcoded short reads are produced that retain the long-range information of the long fragments of the initial DNA extract. Due to the combination of high base pair-level sequence accuracy and long-range information, 10X/Illumina data therefore support excellent SNP and small indel detection and phasing[13], as well as breakpoint detection of large events in cancer[13,15,16]. For diploid genome reconstruction, 10X developed the de novo assembler, Supernova, which has been shown to produce whole human genome assemblies from 56-fold coverage 10×/Illumina data[17,18].

The application of assembly approaches to human genomes has been limited even though they allow powerful identification of SVs[8,19,20]. Long-read-based assemblies, such as those from PacBio data performed by FALCON-Unzip[21], exhibit respectable contiguity and variant detection but still suffer from high cost[9]. Supernova assemblies based on 10X/Illumina data are less expensive and allow detection of all types of variation but power is limited because a substantial fraction of the genome is not assembled in a diploid state and genotyping error is still high[22]. Overall, cost-effective assembly-based approaches still suffer from incomplete resolution of the diploid genome and limited power of

variant detection in a personal genome. On the other hand, assembly-based approaches have two advantages: detection of variants is greatly simplified to pairwise alignments rather than complicated read-map-based inference, which is particularly challenging for indels; and the detection of sequences not present in the reference.

Compared to reference-based approaches[23], the competitive disadvantage of de novo assembly methods is that they disregard the high information content of the reference. Depending on genetic background, >99% of anybody's two haplomes outside of centromeres and telomeres is identical to the reference, which therefore constitutes a highly accurate scaffold for personal genomes. It stands to reason that, in principle, an assembly-based method that incorporates information from the reference sequence should combine the advantages of both approaches. We were therefore motivated to develop a new approach to accomplish these three goals via a reference-assisted, assembly-based approach: high-quality diploid personal genome reconstruction; accurate detection of SNPs, indels, and SVs; phasing of all types of variants.

Our method, Aquila, makes use of the reference genome by performing local assembly in small chunks separately for each haplotype, yielding a diploid whole genome consisting of local, phased, contigs whose scaffolding is provided by the reference sequence. It then discovers the most important types of variation on the basis of pairwise alignment to the reference, and infers phasing for all types of assembled variants through previous long-range phasing information. We test its performance with six libraries of 10× linked-reads data for NA12878 and NA24385 individuals, which we had previously generated for evaluation of linked-read-based de novo assembly with Supernova[18,22]. We show that Aquila offers excellent small indel and SV detection at virtually no compromise for SNP detection, as well as highly accurate phasing of the vast majority of heterozygous variants, at reasonable reagent and computational costs.

## Results

**Aquila's motivation, architecture, and workflow.** The motivation of Aquila is to generate sufficiently long contigs from each parental haplotype such that variation can be discovered on the basis of pairwise alignment of these contigs to the reference. Contigs on the order of 100 kb (implying ca. 60,000 contig breaks in a diploid 3 Gb genome) are in principle sufficient to discover most variation as long as the vast majority of the genome is recovered in a diploid state. Hence, the contiguity of the assembly is less important than the diploid nature of it. Aquila's use case is distinct from that of de novo assembly, where the luxuries of a high-quality reference sequence and extensive variation benchmarking resources do not exist. Aquila's motivation is to leverage these luxuries in the service of comprehensive variation discovery.

Aquila works on the autosomes and the X chromosome. It consists of four stages (Fig. 1a): Haplotyping and sorting inferred long fragments and their reads into the two parental bins, locally partitioning reads of each bin, assembling each local partition into sequence contigs, and finally variant calling and phasing.

Each stage has been implemented as a specific module (Fig. 1b). In the Haplotyping module, the original long DNA fragments are reconstructed based on barcode-aware alignment of the reads to the reference sequence. In parallel, SNPs are detected based on these same alignments. Fragments are then clustered into either parental bin labeled by each pair of heterozygous SNPs, and a probabilistic model (see the "Methods" section) is applied to exclude clusters caused by sequencing errors. The clusters are then merged into fewer but larger ones by a greedy recursive algorithm, preserving the separation of parental bins. The resulting clusters

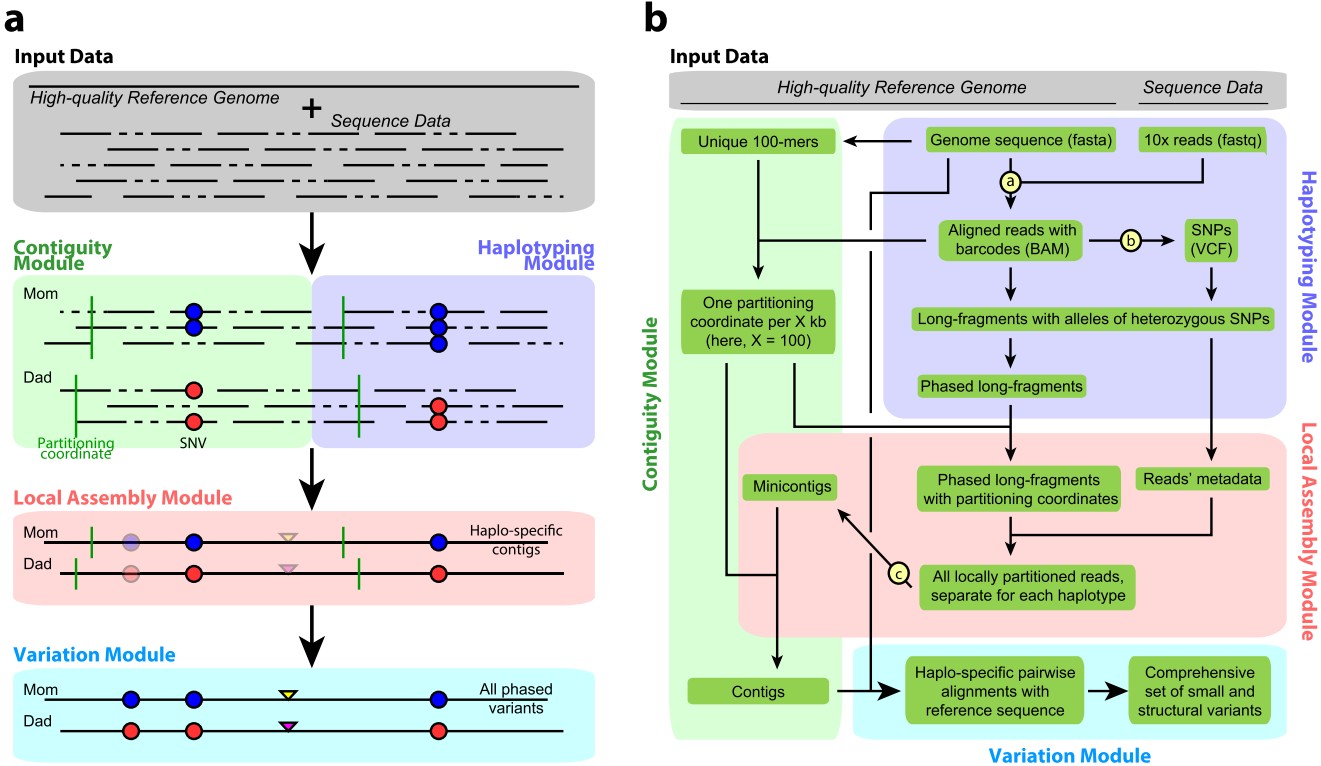

**Fig. 1 Aquila architecture.** Lowercase letters in circles denote existing programs we integrated (a = LongRanger Align, b = FreeBayes, c = SPAdes). **a** Overall architecture. **b** Detailed workflow. Green boxes are data, arrows indicate input and output of a pipeline component. Input data are a high-quality reference genome and 10X-based short reads, each with a barcode (not shown). The Haplotyping module produces phased virtual long fragments (by alignment of reads to the reference, SNP detection, and haplotyping) that become part of each read's record. The Contiguity module produces specific single-base coordinates ('partitioning points') in the genome and in the data, where haplotype blocks and reads are cut at a specific single location, and where subsequently assembled minicontigs are rejoined in the end. The Local Assembly module executes assembly of the reads of a specific region, separately for each parental copy. The variation module then discovers, integrates, and infers the phase of all variation.

contain all reads from a haplotype block of one parent only and are therefore free of allelic variation, greatly simplifying the later assembly steps.

Before assembly is carried out, haplotype blocks whose lengths exceed a user-defined threshold (default = 200 kb) are partitioned into smaller chunks (default = 100 kb) by the Contiguity module. This step is necessary because a considerable fraction of the data is in large phase blocks (up to the length of an entire chromosome arm) and handing all these reads to the assembler would partially defeat Aquila's motivation to sidestep the extreme complexity of whole-genome de novo assembly. Thus, the assembly must be broken down into smaller chunks, preferably in places that are highly likely to produce locally correct assemblies, so that contigs from neighboring sub-assemblies can be spliced back together at the partitioning points with high confidence (see the "Methods" section; Supplementary Fig. 1). The locations of the partitioning points are therefore selected such that (1) they are not within repeats in the reference genome and (2) exhibit the expected read coverage and perfect mapping uniqueness in the abovementioned short-read alignments.

The Assembly module (using SPAdes but any appropriate assembler can be used) produces the complete set of "mini-contigs" from the partitions, which are then spliced together to produce final contigs (Supplementary Fig. 1). While the emphasis in personal genomes is on producing variant calls (see below), we note that traditional short-read-based methods that go directly from read alignment to variant call do not produce contigs as output. The set of final contigs represents a valuable personal resource in the form of a genome sequence.

The Variation module then aligns the assembled contigs to the reference. All of the small indels and SVs that Aquila reports are detected on the basis of these pairwise alignments. SNPs are also detected but this assembly-based set is merged with the initial set of SNPs identified on the basis of the barcode-aware read alignment performed by the Haplotyping module. The last step in the Variation module is the phasing of all discovered heterozygous variants. The final output is a variant call format (VCF) with all phased variation that also includes all detected variants for which the individual is homozygous compared to the reference sequence, and the small fraction of heterozygous variants that could not be phased.

**Characteristics of Aquila assemblies of six libraries.** To explore the performance of Aquila we used six libraries of 10X linked-read sequencing data that were previously generated by us[18,22] from gentle DNA preparations of NA12878 and NA24385 cell lines. The average inferred DNA fragment lengths and their distributions varied among libraries (Supplementary Table 1). We numbered them for each individual according to physical coverage, in ascending order. All of the libraries had approximately 100× Illumina sequencing coverage, except L2 which had 192×. We assembled these libraries with Aquila and compared standard assembly statistics with Supernova 2 assemblies[22] of the same libraries (Table 1, Supplementary Fig. 2). Down-sampling to ~60× was performed for Supernova2 assemblies since that is the maximum coverage it is capable of handling[18]. Contiguity, as measured by Contig N50, was generally >100 kb; NA50, which is

the N50 of contigs after breaking at potential misassemblies by comparison to the reference genome, was in most cases <5% lower, indicating few misassemblies. We note that using the reference to identify potential misassemblies is a conservative approach given that it is not the same individual as the sequenced one. Contiguity generally increased as a function of physical coverage ($C_F$; L5 producing the best result at $C_F = 803\times$ and $C_R = 0.08$), and was greatest with a weighted fragment length of around 150 kb (Supplementary Fig. 2). k-mer spectrum analysis[24] indicates excellent completeness of the diploid nature of the assemblies as well (e.g., Supplementary Fig. 3).

The fraction of the reference genome covered by the assemblies ranged around 95%, indicating that the vast majority of the non-N and not highly repetitive regions of the genome were covered by all libraries assembled by either Aquila or Supernova2. Aquila consistently produced 98% of this fraction in a diploid state, compared to Supernova's 73–78% (Table 1). This key metric indicates that Aquila produces assemblies that have the potential to support diploid variant detection genome-wide.

Aquila also seamlessly supports combining different libraries, if greater contiguity is desired than is achievable with a single library (Supplementary Fig. 4). To illustrate this, we computed the same metrics as for the single libraries for the combination of L5 and L6 from NA24385. Contig N50 increased by 30–40% to

178 kb, the fraction of reference genome assembled rose to 97%, and the diploid fraction reached 99% of the assembled genome. While assembly statistics represent an important facet for evaluation of Aquila, the ultimate metric for the usefulness of the approach is how well it detects genetic variation. We therefore evaluated the assemblies for their ability to support detection of SNPs, small indels, and larger SVs.

**Assembly-based detection of SNPs and small indels.** We first evaluated assembly-based SNP and small-indel (<50 bp) detection by comparing Aquila's calls against the Genome in a Bottle (GiaB) benchmark callsets[25]. The libraries with the best assembly statistics, L3 (from NA12878) and L5 (from NA24385), achieved 97.4% and 97.8% accuracy (F1 metric) for SNPs (Table 2; Supplementary Table 2) and >93% accuracy for the high-confidence set of GiaB small indels (Table 3; Supplementary Table 3), which represents substantial improvement over the indel callset of FreeBayes and Longranger. Genotyping errors for the calls that matched GiaB were 0.14–0.16% for SNPs and 1.61–1.90% for high-confidence small indels. The total numbers of assembly-based SNP calls (Supplementary Table 4) were 3,971,444 (L3) and 3,882,869 (L5), compared to the total numbers of FreeBayes-based calls performed on the barcode-aware read alignments of

---

**Table 1 Assembly metrics for the six libraries we built from NA12878 and NA24385 (L1–L6) and a previously published 10X Genomics library (L7); data was downsampled for Supernova to its acceptable maximum of ca. 60× by its parameter "--maxreads".**

| | Raw coverage C (X) | $C_F$ (X) | $C_R$ (X) | Contig N50 (bp) | Contig NA50 (bp) | Diploid fraction (%) | Genome fraction (%) |
|---|---|---|---|---|---|---|---|
| *NA12878* | | | | | | | |
| L1_Aquila | 103 | 123 | 0.41 | 34,759 | 31,645 | 98.1 | 95.45 |
| L1_Supernova | | | | 58,408 | 57,254 | 73.3 | 93.69 |
| L2_Aquila | 192 | 334 | 0.27 | 94,699 | 91,662 | 96.7 | 94.43 |
| L2_Supernova | | | | 144,298 | 136,253 | 58.9 | 94.81 |
| L3_Aquila | 106 | 958 | 0.07 | 120,963 | 116,438 | 98.7 | 96.17 |
| L3_Supernova | | | | 114,900 | 111,388 | 77.2 | 94.62 |
| L7_Aquila | 78 | 243 | 0.32 | 78,309 | 75,378 | 97.1 | 95.32 |
| *NA24385* | | | | | | | |
| L4_Aquila | 100 | 208 | 0.25 | 96,558 | 93,132 | 98.2 | 96.98 |
| L4_Supernova | | | | 99,516 | 96,510 | 79.2 | 95.01 |
| L5_Aquila | 100 | 803 | 0.08 | 135,115 | 130,176 | 98.8 | 96.77 |
| L5_Supernova | | | | 129,195 | 125,339 | 78.1 | 94.86 |
| L6_Aquila | 100 | 1504 | 0.05 | 125,064 | 120,490 | 98.7 | 96.62 |
| L6_Supernova | | | | 101,236 | 98,364 | 73.4 | 94.69 |
| L5+L6_Aquila | 200 | 2307 | 0.07 | 177,974 | 172,358 | 99.1 | 97.06 |

$C_F$ physical ('fragment') coverage; $C_R$ read coverage; $C$ raw coverage $\geq C_F \times C_R$. Genome fraction, percentage of reference genome that is covered by the assembly. Diploid fraction, percentage of genome fraction that is covered by exactly two parental contigs. L5 + L6 describes performance for a simple combination of the data from libraries 5 and 6. The statistics of the female individual, NA12878, include the X chromosome; those of the male individual, NA23485, include the X chromosome but not the Y chromosome.

---

**Table 2 Accuracy of SNP calling, comparing assembly-based calling with two mapping-based approaches on the same libraries' linked read data, one each from NA12878 (L3) and NA24385 (L5).**

| SNPs | | True positives | False negatives | False positives | Genotype mismatch | Precision | Recall | F1 |
|---|---|---|---|---|---|---|---|---|
| L3 (NA 12878) | Aquila (assembly only) | 3,004,501 | 38,282 | 124,074 | 4730 | 0.960 | 0.987 | 0.974 |
| | FreeBayes | 3,037,504 | 5279 | 54,088 | 3501 | 0.983 | 0.998 | 0.990 |
| | Longranger | 3,040,701 | 2082 | 105,854 | 1621 | 0.966 | 0.999 | 0.983 |
| L5 (NA 24385) | Aquila (assembly only) | 2,989,567 | 39,785 | 93,195 | 4157 | 0.970 | 0.987 | 0.978 |
| | FreeBayes | 3,021,814 | 7544 | 48,477 | 3899 | 0.984 | 0.998 | 0.991 |
| | Longranger | 3,026,384 | 2974 | 104,879 | 1799 | 0.967 | 0.999 | 0.983 |
| L5 + L6 (NA 24385) | Aquila | 2,971,237 | 58,120 | 81,926 | 18,856 | 0.973 | 0.981 | 0.977 |

The benchmark is GiaB v3.3.2. Variant counts and performance scores were generated by RTGtools/hap.py. Longranger calls were executed with "-vcmode = gatk". For final SNP calling, Aquila combines mapping-based calls from FreeBayes with its assembly-based calls. L5 + L6 can be achieved by Aquila through a multiple-library assembly mode, which is not applicable for other tools.

---

**Table 3 Accuracy of small indel calling, comparing assembly-based calling with two mapping-based approaches on the same libraries' linked read data, one each from NA12878 (L3) and NA24385 (L5).**

| Small indels | | True positives | False negatives | False positives | Genotype Mismatch | Precision | Recall | F1 |
|---|---|---|---|---|---|---|---|---|
| L3 (NA 12878) | Aquila | 499,301 | 32,081 | 40,292 | 9493 | 0.925 | 0.940 | 0.932 |
| | FreeBayes | 419,344 | 80,354 | 45,977 | 39,636 | 0.903 | 0.839 | 0.870 |
| | Longranger | 463,732 | 35,966 | 88,431 | 22,513 | 0.843 | 0.928 | 0.883 |
| L5 (NA 24385) | Aquila | 476,139 | 26,914 | 35,315 | 7986 | 0.931 | 0.946 | 0.939 |
| | FreeBayes | 400,440 | 75,170 | 41,475 | 36,293 | 0.908 | 0.842 | 0.874 |
| | Longranger | 443,107 | 32,504 | 81,044 | 19,720 | 0.848 | 0.932 | 0.888 |
| L5+L6 (NA 24385) | Aquila | 473,895 | 29,158 | 15,724 | 5335 | 0.968 | 0.942 | 0.955 |

The benchmark is GiaB v3.3.2 within the high-confidence regions. L5 + L6 can be achieved by Aquila through a multiple-library assembly mode, which is not applicable for other tools.

3,949,721 (L3) and 3,961,684 (L5). Numbers of heterozygotes or homozygotes are also comparable between the two approaches (Supplementary Table 4). We note that Aquila produces numbers of assembly-based SNP calls for these two individuals that are consistent with those previously produced from standard short-fragment Illumina libraries[26–28].

Compared to the GiaB small indel callset, Aquila produces considerably more calls (e.g., 1,007,313 in L3 vs. GiaB's 531,382; Supplementary Table 5). This difference is due to the current incompleteness of the GiaB callset especially among longer small indels, as well as a false-positive rate of the Aquila calls that we cannot rigorously estimate outside of the GiaB regions. The size distribution of Aquila's small indels matches the size distribution of the GiaB calls very closely, exhibiting the same 2 bp periodicity such that insertions or deletions of an even length are more common than those that are one base longer or shorter (Supplementary Fig. 5). At lengths above 30 bp, where GiaB has very few calls, the Aquila calls continue to exhibit this pattern. The correlations between the Aquila and GiaB distributions of the 1-49 bp indel calls are $R^2 = 0.997$ and 0.998 of the raw counts, and 0.930 and 0.951 for the natural log of counts, for insertions and deletions, respectively.

**Assembly-based detection of SVs 50 bp and greater.** Aquila calls ca. 17,000 deletions and ca. 6000 insertions 50 bp and greater in each high quality library (L2, 3, 5, and 6; Supplementary Table 6). The size distributions follow the expected exponential distribution with a peak at ca. 330 bp, which is caused by full-length or nearly full-length Alu elements (Fig. 2a). The number of calls is comparable but consistently lower than recent estimates from a comprehensive study that focused exclusively on SVs in a cohort of long-read sequenced individuals, including two haploid samples, in which purpose-driven approaches were applied to achieve high sensitivity of detection of shared SVs[29,30]. We do not expect to reach the same level of detection in a single personal genome without the benefit of leveraging several individuals to inform discovery, but we are able to apply several metrics to characterize Aquila's SV calls (Fig. 2b–g).

We initially used a combination of three strategies to validate the SV calls in both individuals: First, for both individuals, PacBio data exist that we applied with svviz2[31] to test Aquila's calls with another data type; second, the two individuals are expected to share a large fraction of SVs, so we performed simple comparisons of the call sets; third, we aligned the SVs and flanking sequence to two high-quality ape genomes (Chimp and Orang). Given the complexity of SV calling we expect to have both false negative and false positive calls, which is underscored by the fact that different libraries from the same individual produce strongly overlapping but not fully identical call sets. The calls that are shared between libraries validate

(by at least one of the abovementioned three metrics) at 92% for NA12878 and 82% for NA24385 (Fig. 2b), whereas calls unique to each library validate at lower rates (40–52%). This shows that we do not have perfect sensitivity (because there are many validated calls unique to a single library) and, conversely, that the nonvalidated set of calls unique to a library likely contains false positives. The number of called SVs exhibit the expected dropoff as a function of derived allele frequency, as estimated in this sample of five haplomes (two individual plus reference; Supplementary Fig. 6).

The number of SVs validated by svviz2/PacBio is higher than those validated by the other two approaches (Fig. 2c), since the PacBio data is from the same individual as the respective assembly. However, there are many SVs that are not validated by PacBio but that are also called in the other individual or are validated by comparison to ape sequences. This effect is greater for deletion calls than for insertion calls (Fig. 2c), which suggests that insertions may be called at higher specificity and lower sensitivity than deletions. This interpretation is consistent with a predicted shortcoming of the current implementation of Aquila, which because of its reliance on the reference sequence to identify reads for assembly has decreasing power to assemble insertions as their size increases.

We also assessed the consistency of the breakpoints of the calls that are shared between the two individuals (Fig. 2d, e). We binned the SVs based on the size differences between the calls in NA12878 and NA24385, as a function of validation by the other two approaches (Fig. 2d) or as a function of sequence type (Fig. 2e). Overall, the vast majority of calls have precisely the same breakpoints in both individuals. Deletions or validated calls have better precision than insertions or nonvalidated calls, and SVs in repeats have worse precision than nonrepetitive sequences.

**Genome in a bottle benchmark comparison.** During the course of this work, the first GiaB SV benchmark, v0.6, was released. It is based on the HG19 reference sequence and is specific to NA24385, with 9397 SVs >50 bp in the call set. Accuracy (F1 metric) of Aquila calls, assuming all GiaB calls are correct, ranges from 52% to as high as 87% depending on the type of variant and the location in which it occurs (Table 4). This compares favorably with results obtained from Supernova assemblies of the same libraries, and the callset from Longranger: Aquila's recall is 80%, compared to Longranger's (40%) and Supernova's (24–53%)[12,22]. In general, Aquila's recall performance is better than precision, which is likely due to a combination of a rate of actual false positives called by Aquila and an unknown number of false negatives in the GiaB callset.

**Inference of derived alleles.** SV calls are labeled with respect to the reference sequence as 'insertion' or 'deletion', but the

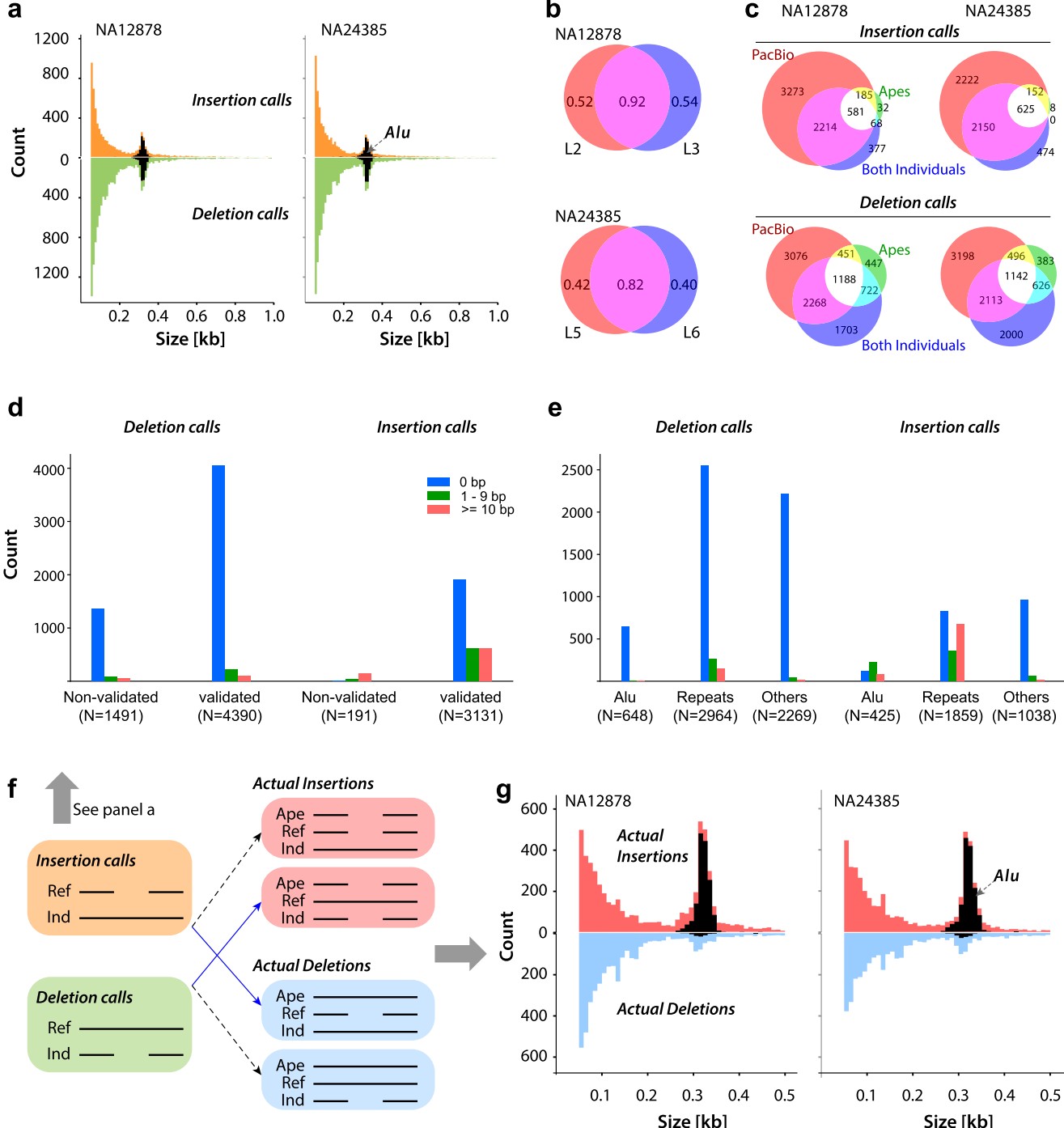

**Fig. 2 Characteristics and validation statistics of SV calls by Aquila. a** Size frequency distributions of insertion calls and deletion calls in both individuals (L3 for NA12878 and L5 for NA24385). Black areas represent indels of which at least 80% are a close match to Alu-element consensus sequences. **b** Call validation rates by three validation strategies of two libraries (L2 and L3; L5 and L6) per individual; SVs called in both libraries are in the overlap, flanked by SVs unique to each library. **c** Overlap analysis and comparison of three validation strategies, by call and individual; numbers inside the Venn diagrams are counts of SVs. SVs are validated by: PacBio data from the same individual, (PacBio); the other individual (Both Individuals); in the chimp or orang genome (Apes). Overlaps represent two or more of these criteria fulfilled. **d**, **e** Comparative precision of SVs present in both individuals, as a function of validation by three validation strategies (**d**) or sequence class (**e**). Bar graphs depict counts of SVs that have precisely the same breakpoint coordinates in both individuals (0 bp), that differ by <10 (1–9 bp), or that differ by 10 or more (≥0 bp). "Repeats" class includes simple sequence and tandem repeats but not mobile elements; "Other" class includes all SVs that do not overlap more than 80% with Alus and are not part of the Repeats class. **f** Inference of actual molecular mechanism that produced the SV by expanding the alignment between the reference sequence (Ref) and the alternate allele call from the Individual (Ind) to include chimp or orang sequences; the sequence (reference or alternate) that matches the ape is the ancestral allele. "Actual insertion" and "Actual deletion" refer to the molecular mechanism that produced the derived allele. Approximately 45% of deletion and 24% of insertion calls are thus 'inverted' (blue arrows). **g** Size frequency distributions of actual insertions and actual deletions in both individuals. Black areas represent indels of which at least 80% are a close match to the Alu-element consensus sequence. The peak at around 330 base pairs captures nearly all Alu SVs.

**Table 4 Overall benchmarks of SVs from L5, L6, and L5 + L6 by Aquila.**

| SVs | | True positives (not-in/in break) | False negatives (not-in/in break) | False positives (not-in/in break) | Precision (not in break) | Recall (not in break) | F1 (not in break) |
|---|---|---|---|---|---|---|---|
| L5 | Total | 6197 (5631/566) | 3200 (1424/1776) | 7776 (6181/1595) | 0.443 (0.477) | 0.659 (0.798) | 0.530 (0.597) |
| | No TR > 100 bp | 3900 (3578/322) | 1636 (445/1191) | 3806 (3046/760) | 0.506 (0.540) | 0.704 (0.889) | 0.589 (0.672) |
| | No TR any size | 1546 (1429/117) | 916 (297/619) | 466 (398/68) | 0.768 (0.782) | 0.628 (0.828) | 0.691 (0.804) |
| L6 | Total | 6086 (5551/535) | 3311 (1460/1851) | 8021 (6264/1757) | (0.470) | 0.648 (0.792) | 0.518 (0.590) |
| | No TR > 100 bp | 3830 (3515/315) | 1706 (471/1235) | 3889 (3036/853) | 0.496 (0.537) | 0.691 (0.882) | 0.577 (0.667) |
| | no TR any size | 1527 (1428/99) | 935 (301/634) | 430 (365/65) | 0.780 (0.796) | 0.620 (0.826) | 0.691 (0.811) |
| L5+L6 | Total | 6434 (5983/451) | 2963 (1416/1547) | 5577 (4404/1173) | 0.536 (0.576) | 0.685 (0.809) | 0.601 (0.673) |
| | No TR > 100 bp | 4065 (3803/262) | 1471 (456/1015) | 2848 (2256/592) | 0.588 (0.628) | 0.717 (0.893) | 0.646 (0.737) |
| | No TR any size | 1671 (1574/97) | 791 (294/497) | 223 (185/38) | (0.895) | 0.662 (0.842) | 0.756 (0.867) |

SVs were called from the NA24385 assemblies and compared to GIAB NIST_SVs_Tier1_v0.6 (bed file HG002_SVs_Tier1_v0.6_chr.bed). SV counts and performance scores were generated by Truvari (parameters -p 0.1 -P 0.1 -r 200 --passonly). No TR > 100, without SVs if at least 20% of the reference bases are tandem repeats at least 100 bp long; No TR any size, without SVs if at least 20% of the reference bases are in tandem repeats of any size. Numbers in parentheses (not-in/in break) are counts of SVs outside or within assembly breaks.

molecular mechanism that generated the SV may be the opposite of the call because the reference sequence is a random sample of ancestral and derived alleles from the population. Where the reference carries a derived allele, the actual molecular mechanism that generated the SV is the opposite of the call because the assembly carries the ancestral allele. For those SVs with alignments to ape genomes, the SV allele matching the ape sequence is highly likely to represent the ancestral state: when the ape sequence matches the reference, the alternate allele (which was the 'SV call' in the individual) is derived; when the ape sequence matches the alternate allele, the reference allele is derived and the alternate allele call from the individual is ancestral (Fig. 2). Classifying the SVs accordingly ("actual" insertion or deletion) causes a striking shift in the size distributions of insertions and deletions in both individuals (Fig. 2a, g), in which the vast majority of SVs that overlap Alu repeat sequences are now revealed to be actual insertions (Fig. 2g). This provides empirical evidence that the classification into ancestral and derived alleles is largely correct, as Alus are known to insert as full length sequences, whereas partial Alus are degenerate copies that arise later in evolution from deletions whose breakpoints can be anywhere in the element.

**Genome-wide distribution and phasing of all variation**. To interrelate the different types of variation detected by Aquila and to ask whether there were any obvious biases we divided the genome into bins of 250 kb and quantified contig density and variation content by genotype from the assembly of L3. Contig density, which over the vast majority of the genome is exactly 2 because of the diploid nature of the assemblies (Fig. 3a), does not correlate with any variation and only weakly with repeat content. Repeat content correlates weakly with the number of SVs. As expected, numbers of SNPs and very small indels correlate strongly when the genotype is the same (heterozygous or homozygous), and weakly with larger small indels, which in turn correlate weakly with SVs (Fig. 3b). Overall, the correlation patterns do not reveal any large-scale biases in variation discovery. We also note that the fraction of variants that are heterozygous varies over a narrow range across all types and sizes of detected variation (Supplementary Fig. 7), again revealing no obvious biases. Phase blocks are very long (Fig. 3a) and allow phasing of coding variants within genes (e.g., Fig. 3c).

The last step in Aquila's pipeline is a final integrative phasing of all of the discovered heterozygous variation on the basis of the phase blocks obtained with heterozygous SNPs in the Haplotyping module (Table 5). Depending on the library, between ca 1.7 and 2 million SNPs were initially phased (Supplementary Table 7). Because the parental genotypes are known for the two individuals we could quantify the phasing error, which is dominated by switch errors that involve a single, presumably incorrectly genotyped, SNP (Table 5). Long switch errors are quite rare, comparing favorably with previous work using 10X data and a variety of phasing algorithms[32,33]. Because SNPs are the densest type of polymorphism in human genomes, phasing other variants on the basis of these is feasible. Instead of probabilistic imputation, however, Aquila performs straightforward inference by matching the assembly-based SNP calls with those of the Haplotyping module and then simply inferring the correct phase (Supplementary Fig. 8). In total, Aquila added ca. 1 million heterozygous variants, including 0.5–0.8 million previously unphased SNPs, ca. 0.5 million small indels, and ca. 10,000 heterozygous SVs in the best four libraries (Supplementary Table 7). Finally, major histocompatibility complex (MHC) genotypes as assessed against the NA12878 benchmark are highly accurate (Supplementary Tables 8 and 9).

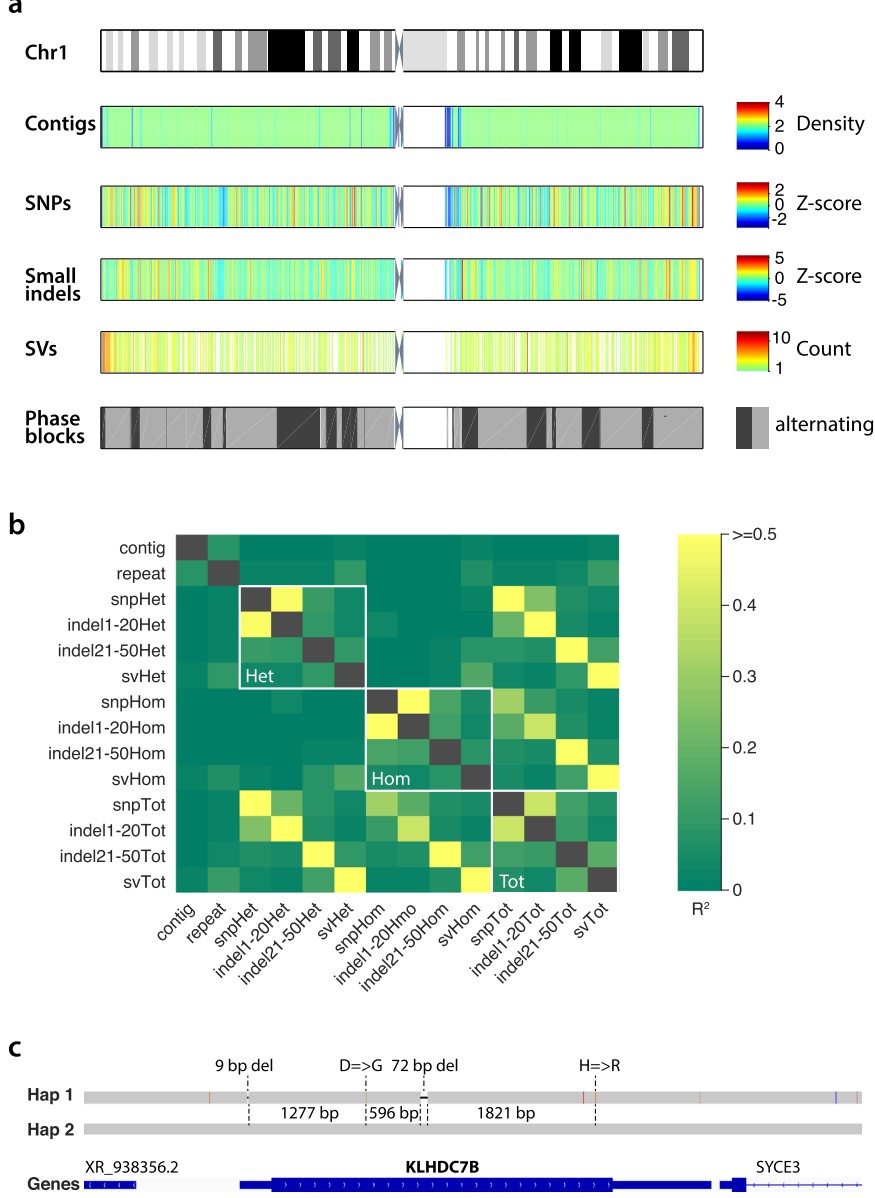

**Fig. 3 Local distribution of all types of variation detected, aggregated across 250 kb intervals. a** Top, ideogram of Chromosome 1 showing metaphase banding pattern. Tracks below are assembly and variation features where white represents no data. Contigs, average contig coverage. SNPs, $Z$-score of SNP number. Small indels, $Z$-score of the number of indels 1–50 bp. SVs, number of SVs $\geq$ 50 bp in each 250 kb bin. Phase blocks, each phase block is a gray rectangle, with alternating light and dark indicating neighboring phase blocks. **b** Genome-wide correlation ($R^2$) among all pairs of variation types by genotype, contig density, and repeat density. **c** Example of phased exonic variants in cis within the KLHDC7B gene, where Haplotype 1 carries the alternate alleles and Haplotype 2 carries the reference alleles.

## Discussion

We introduce a new method, Aquila, which uses a high-quality reference sequence to perform assembly in small chunks for both haplotypes, producing a diploid personal genome sequence from a single data type, Illumina/10X linked reads. We test Aquila's performance on several libraries from two standard individuals and show that it produces high-quality diploid assemblies. The assemblies enable comprehensive discovery of SNPs, small indels, and SVs, on the basis of pairwise alignments to the reference genome. We show that Aquila produces overall better results than de-novo assembly from Supernova[22], for all types of variants. Accuracy of variant discovery, as evaluated against GiaB benchmark sets, retains some characteristics of standard Illumina short-fragment sequencing, with higher accuracy for smaller variants

than for longer ones. Long-range phasing is highly accurate with low switching error.

Aquila effectively leverages the strengths of 10X/Illumina sequencing to enable comprehensive variation discovery and phasing in personal genomes. Compared to standard Illumina sequencing, the increased cost of the 10X sample prep and the generation of deeper sequence data is justified by (1) the much greater power to detect insertions and deletions, and (2) the genome-wide phasing of all heterozygous variants (Fig. 3c). Compared to ONT and standard PacBio, the superior base-pair level accuracy of Illumina sequencing ensures more accurate SNP and small indel detection as well as SV breakpoint determination; and while PacBio HiFi enables highly accurate genome sequencing[12], it remains to be seen whether it performs substantially

**Table 5 Phasing information and accuracy for each library from Aquila versus Longranger.**

| | Method | $W_{\mu_{FL}}$ (kb) | PB N50 (Mb) | Max. PB (Mb) | Long errors | Mismatch errors | Phased in GiaB | Long error (%) | Mismatch error (%) |
|---|---|---|---|---|---|---|---|---|---|
| L1 | Aquila | 304.3 | 25.1 | 104.5 | 453 | 1492 | 1,613,542 | 0.02 | 0.16 |
| | Longranger | | 12.7 | 49.3 | 1092 | 2860 | 1,862,855 | 0.04 | 0.19 |
| | *HapCUT2* | | 58.1 | 131.6 | 565 | 2336 | 1,748,918 | 0.02 | 0.19 |
| L2 | Aquila | 41.1 | 0.6 | 4.9 | 486 | 1532 | 1,697,703 | 0.02 | 0.15 |
| | Longranger | | N/A | N/A | N/A | N/A | N/A | N/A | N/A |
| | HapCUT2 | | N/A | N/A | N/A | N/A | N/A | N/A | N/A |
| L3 | Aquila | 214.5 | 12.7 | 57.5 | 499 | 1481 | 1,323,959 | 0.03 | 0.18 |
| | Longranger | | 12.5 | 57.7 | 1124 | 2679 | 1,868,719 | 0.04 | 0.20 |
| | HapCUT2 | | 55.4 | 121.5 | 451 | 2216 | 1,763,129 | 0.02 | 0.18 |
| L4 | Aquila | 267.4 | 22.1 | 104.5 | 608 | 2182 | 1,313,659 | 0.03 | 0.23 |
| | Longranger | | 17.2 | 61.3 | 1638 | 5345 | 1,583,816 | 0.06 | 0.37 |
| | HapCUT2 | | N/A | N/A | N/A | N/A | N/A | N/A | N/A |
| L5 | Aquila | 151.7 | 15.8 | 62.4 | 719 | 2388 | 1,342,247 | 0.04 | 0.24 |
| | Longranger | | 13.4 | 46.8 | 1682 | 5171 | 1,586,187 | 0.06 | 0.37 |
| | HapCUT2 | | N/A | N/A | N/A | N/A | N/A | N/A | N/A |
| L6 | Aquila | 216.9 | 10.6 | 63.6 | 641 | 2218 | 1,147,993 | 0.04 | 0.26 |
| | Longranger | | N/A | N/A | N/A | N/A | N/A | N/A | N/A |
| | HapCUT2 | | N/A | N/A | N/A | N/A | N/A | N/A | N/A |

*W* weighted fragment length of the library, *PB N50* phase blocks length N50, *Max. PB* maximum phase block length, *Phased in GiaB* number of phased SNPs overlapping with callset v3.3.2, *N/A* not applicable because Longranger could not complete runs in wgs mode.

better than 10X/Illumina/Aquila to justify the additional cost, particularly for cohort studies. Compared to ensemble approaches that computationally integrate multiple data types with complementary strengths (for example, combining Illumina, long-read, and BioNano data), 10X/Illumina/Aquila is far less complex to manage in the laboratory and offers a simpler computational approach.

In the ecosystem of solutions for human whole genome sequencing, 10X/Illumina/Aquila therefore fills what we believe is the most important niche: a diploid and phased personal genome for accurate and comprehensive discovery of SNPs, small indels, and SVs in all but the most complicated regions of the human genome. It represents the first generation of approaches that drive toward laboratory and computational efficiency and simplicity by using a single data type and leveraging the considerable amount of information present in the human reference sequence. Until de novo assembly on the basis of highly accurate very-long-read data is shown to be cost-effective, reference-assisted approaches that partition the genome into smaller assembly problems are likely to prevail.

Further improvements of the approach we take here fall into two categories: those for which the nature of current linked reads data is inherently limited and will require technological advances and those in which future implementations of Aquila will produce better results. For example, it is unlikely that linked reads data will support assembly and resolution of recent segmental duplications or long repetitive sequences. However, the current dropoff in sensitivity to detect insertions beyond 500 bases will be addressed by improving the inclusion of ambiguously mapping reads into the assembly process. Similarly, improvements to assembly contiguity will increase accuracy of variants in repetitive sequences, which are currently enriched in assembly breaks. Future improvements will also center on better detection of long insertions and contig breakpoint assembly. Although developed for 10X/Illumina data, Aquila's architecture may be used in the future for computational approaches that use data from other sources. In its current form, it is already applicable to any study that requires better indel detection than what is achievable with standard Illumina sequencing.

## Methods

Aquila is organized in four conceptual modules that correspond to the following python steps (Fig. 1): Aquila_step1.py: Haplotyping module + contiguity module (first part: partitioning). Aquila_step2.py: Local assembly module + contiguity module (second part: concatenation). Variation Module: Aquila_assembly_based_variants_call.py and Aquila_phasing_all_variants.py.

**Pruning unreliable variants (haplotyping module).** Accurate haplotyping requires filtering out incorrectly genotyped variants and false positives due to sequencing error. We performed an empirical analysis for 10X data to investigate the alternate allele frequency ($R_{alt/ref}$) and coverage per variant ($d_{var}$) that could be used as metrics to find erroneous calls. Allele frequency ($R_{alt/ref} \geq 0.25$) was used for a cutoff, and a 2-tailed percentile cutoff was used for coverage per variant (10% *avg_cov $\leq d_{var} \leq$ 90%*avg_cov, avg_cov: average read coverage per variant). The haplotyping algorithm was further improved by sacrificing a small amount of low-confidence heterozygous variants. SNP quality (13 by default) is the final free parameter used to prune variants.

**Inference and phasing of original long-fragments (haplotyping module).** All DNA fragments are first reconstructed by aligning short reads to the human genome reference (Hg38) by a barcode-aware alignment strategy (Longranger align, https://support.10xgenomics.com/genome-exome/software/pipelines/latest/installation). Aquila then sorts all reads by barcodes and positions, and collects the reads with the same barcode to reconstruct each fragment. There is a threshold to differentiate two molecules with the same barcode when the distance between two successive reads with the same barcode is larger than 50 kb (50 kb by default, free parameter). After reconstructing all fragments, Aquila assigns the alleles of heterozygous SNPs to each fragment by scanning the reads belonging to each fragment and comparing to a VCF file generated by FreeBayes. At a heterozygous locus "0" is the reference allele and "1" is the alternate allele.

For each pair of heterozygous variants, if the even parity was correct where one haplotype supported "00", and the other haplotype supported "11", the odd parity must then have been caused by a sequence error with some fragments supporting "01", and other fragments supporting "10", and vice versa. In rare scenarios, the fragment could have two sequencing errors when the even/odd parity was correct, but the fragment supported the complementary haplotype (e.g., the haplotype is "00", and the fragment supports "11"). For each fragment with at least two heterozygous SNPs, Aquila records all neighboring pairs of heterozygous variants. It then applies a Bayesian model (see below) to evaluate if even or odd parity is correct, and the clusters with the parity caused by sequencing error are excluded from further steps. Importantly, the excluded clusters are due to the variants caused by sequencing errors, not the molecules themselves, which means if these molecules still contain other pairs of heterozygous variants with consistent haplotype with the correct parity, they are still used for haplotyping.

Aquila then performs a recursive clustering algorithm in two haplotypes to aggregate bigger clusters/phase blocks. Two clusters are merged if the number of molecules in both of them supporting the same haplotype exceeds a threshold. This threshold is set to 3 by default, which corresponds to a merging error

percentage $\leq ((1-p_1)(1-p_2))^3$, for each pair of variants, if each variant matched the true variant with probability $p_1$ and $p_2$, respectively. Aquila sorts all pairs of clusters by the positions of reads of all molecules in each cluster. When two locally successive clusters are merged into one single cluster, the corresponding clusters of the other haplotype are merged too. The resulting pairs of clusters are sorted again for the next iteration. The sorting algorithm complexity is $\sim O(N_{var} \log N_{var})$, where $N_{var}$ is the total number of heterozygous variants. Aquila performs clustering recursively until no more clusters can be merged based on the supporting threshold.

The result of this step are pairs of clusters where each pair corresponds to one diploid phase block. For each phase block, Aquila then performs haplotype construction and extension when each heterozygous variant is supported by all the molecules that cover it. When there are multiple molecules supporting inconsistent genotypes for a variant, that variant is excluded from further steps. To further extend the phase blocks, Aquila similarly performs recursive clustering when two phase blocks have a number of overlapping variants greater than a certain threshold. The threshold is set to 5 by default so that the merging error due to sequencing error $p$ is $\leq p^5$. When no more phase blocks can be merged the process has converged.

In the final step, all the molecules/fragments with at least two heterozygous variants are assigned to a phase block based on the variants in the final phase block, and a maximum-likelihood estimation is applied. Given a haplotype $H$ and a molecule $M$, we apply the theta function $\theta(H_i, M_i) = 1$ if $H_i = M_i$ and 0 otherwise. Given $p_i$, the probability the allele call at variant $i$ in molecule $M$ is correct, the likelihood of observing molecule $M$ is

$$p(M|p, H) = \Pi\theta(H_i, M_i)p_i + (1 - \theta(H_i, M_i))(1 - p_i) \quad (1)$$

Similarly, given the complementary haplotype $H_c$ and molecule $M$, a theta function gets the value $\theta(H_{ci}, M_i) = 1$ if $H_{ci} = M_i$ and 0 otherwise. The likelihood of observing molecule $M$ is: $p(M|p, H_c) = \Pi\theta(H_{ci}, M_i)(1 - p_i) + (1 - \theta(H_{ci}, M_i))p_i$. To assign the final phase block to each molecule $M$, Aquila needs to find the haplotype $j$ in the final phase blocks that meets $\mathrm{argmax}_j \left( p(M|p, H^j) - p\left(M|p, H_c^j\right) \right)$.

**Probability model to determine correct joint key (haplotyping module).** At each position, the genome has two complementary keys matching the true two haplotypes: 00, 11 (even parity) or 01, 10 (odd parity). Each variant matches the true variant with probability $p_1$ and $p_2$, respectively. The probability that a sequence key will have the correct parity is $p_c = p_1p_2 + (1-p_1)(1-p_2)$, since both variants could match the true variants or both variants are called wrong. Let $N$ be the number of sequences observed that have both variants, and $k$ be number of sequences with key of even parity, and $(N-k)$ be the number of sequences with key of odd parity. Using Bayes' theorem to test $P(B|A)$:

Where $A$: $k$ out of $N$ molecules have keys with even parity, $B$: true key is even parity.

$$P(B|A) = \frac{P(A|B)P(B)}{P(A)} \quad (2)$$

Aquila will accept even parity is correct if $P(B|A)$ exceeds a significance level (e.g. >0.99), where:

$P(B|A)$ is the probability of the true key being even parity given $k$ out of $N$ molecules have keys with even parity.

$P(A|B)$ is the probability of $k$ out of $N$ molecules having keys with even parity given the true key is even parity, which is $P(A|B) = \binom{N}{k} \cdot p_c^k (1 - p_c)^{N-k}$.

$P(A)$ is the probability of $k$ out of $N$ molecules have keys with even parity, which is

$$P(A) = \binom{N}{k} / \left( \binom{N}{1} + \binom{N}{2} + \dots + \binom{N}{k} + \dots + \binom{N}{N} \right) = \binom{N}{k} / 2^N.$$

$P(B)$ is the probability of even parity being correct, $P(B) = 1$.

**High-confidence partitioning point profile generation (assembly module).** Large phase blocks (free parameter:-- block_threshold = 200 kb by default) are cut into multiple small chunks of a specific length (free parameter: --block_len_use = 100 kb by default) based on a high-confidence partitioning point profile. This is done to make assembly faster and more tractable, by avoiding too many reads being given to the assembler. This profile is generated based on three criteria: 1. Expected reads coverage ($C$), 2. Expected physical coverage ($C_F$), 3. 100-mer uniqueness. Read depth and fragment/physical coverage for each position is calculated after reconstructing all the fragments. The 100-mer uniqueness file[34] for hg38 processed and included in Aquila, was downloaded from http://hgdownload. soe.ucsc.edu/gbdb/hg38/hoffmanMappability/k100.Unique.Mappability.bb

Each locus/position is defined to be a high-confidence partitioning point if $C_{(partitioning\ point)} > C_{(average)}*0.8$, $C_{F(partitioning\ point)} > C_{F\ (average)}*0.8$, and locus $\epsilon$ 100-mer uniqueness. Aquila uses these high-confidence partitioning points in the profile as reference points to partition reads before assembly (see next section) and later to reconnect the resulting mini-contigs into contigs (Supplementary Fig. 1).

**Local assembly within small diploid chunks and stitching contigs (assembly and contiguity modules).** For each phase block (Supplementary Fig. 1), Aquila records its original long molecules and their corresponding short reads. Large phase blocks are cut into small chunks (see previous step) to perform local assembly with SPAdes[35] within each phase block for both haplotypes separately. (SPAdes is included in the Aquila package.) Those resulting minicontigs from neighboring small chunks that are bounded by the same partitioning point are concatenated. 99% of partitioning points met this criterion. For each concatenating iteration, the previous concatenated contig is used for the next iteration of concatenation. At the end of this step, Aquila has generated contigs for both haplotypes in each original phase block. The algorithm complexity is $\sim N_{chunks}O(T_{onechunk})$, where $N_{chunks}$ is the number of small chunks and $T_{onechunk}$ is the time for finishing assembly of one small chunk.

**Variation detection for assembled contigs (variation module).** To generate SNPs, small Indel, and SV calls from the de novo assemblies, Aquila uses the contig file of the haplotype 1 and haplotype 2 of each phase block. Minimap2[36] and paftools (https://github.com/lh3/minimap2/tree/master/misc) are integrated and applied to call variants from each (haploid) contig ("-cx asm5 –cs" is applied for minimap2, and "-l 1 -L 1 -q 20" is applied to paftools). For contig alignments and variant discovery, contigs ≤ 1 kb are filtered out and mapping quality ≥ 20 is chosen to produce variant candidates. Finally, to generate SNPs, if both haplotypes cover the alternate allele, it is defined as homozygous; if one haplotype covers the reference allele and the other haplotype covers the alternate allele, it is defined as heterozygous.

To generate small Indels and SVs, variant candidates from each haplotype are compared against each other to infer zygosity. To achieve that, heterozygous variants are defined if one haploid assembly contains alternate allele(s) and the other haploid assembly contains reference allele(s). Homozygous variants are defined if both haploid assemblies contain alternate allele(s). For compound indel/ SV, we split them into two heterozygous variants. Check "--all_regions_flag = 1" for "Aquila_assembly_based_variants_call.py" in GitHub to perform these analyses.

**Phasing inference (variation module).** The initial phased SNPs from the Haplotyping module provide the scaffold on which all other heterozygous variants that are discovered by the Variation module are phased (Supplementary Fig. 6). For example, consider the case of one assembled SNP in a phase block, "G|A" or "1|0" (where "A" is the reference allele, and "G" is the alternate allele), and the other neighboring assembled SNP in the same phase block, "C|T" or "0|1" (where "C" is the reference allele, and "T" is the alternate allele): these two phased SNPs have the same genotype and phase in a phase block from the haplotyping module. Therefore, Aquila places the SV into the haplotype that is in the same phase. This is done for all heterozygous variants discovered by the Variation module.

**Contig quality assessment.** We used QUAST[37] to generate various assembly metrics such as N50 and NA50. --extensive-mis-size 1000 was applied as the lower threshold of the relocation size.

**Aquila computation cost.** Aquila's four modules vary in their memory requirements and are flexible with respect to computing architecture, allowing jobs to be run in parallel on a cluster or serially in a large-memory machine. Generally the run time is considerably longer on the large memory machine. On a cluster, jobs are parallelized by chromosome or a combination of chromosomes that minimizes the number of nodes required. Modules 1–3 use 23, and module 4 uses 10 h of wall clock time, respectively, on a current-generation standard compute cluster (23 nodes with 128 Gb RAM each, 2 CPUs each, 10-cores per CPU, i.e. 240 cores). Check tables of computation cost in GitHub for more details.

**Aquila output.** Aquila outputs an overall contig file "Aquila_contig.fasta", and separately for each chromosome "Aquila_Contig_chr*.fasta". After performing assembly-based variant calling in the Variation Module, one contig file for each haplotype is generated: "Aquila_Contig_chr*_hp1.fasta" and "Aquila_-Contig_chr*_hp2.fasta". For each contig, the header (for example ">36_PS39049620:39149620_hp1") follows the format contig number ("36"), phase block start coordinate ("PS39049620"), phase block end coordinate (":39149620"), and haplotype number ("hp1"). Within the same phase block, the haplotype number "hp1" and "hp2" are arbitrary for maternal and paternal haplotypes. For some contigs from large phase blocks, the headers are much longer and complex, for an instance, ">56432_PS176969599:181362362_hp1_ mer-ge177969599:178064599_hp1-177869599:177969599_hp1". "56" denotes contig number, "176969599" denotes the start coordinate of the final big phase block, "181362362" denotes the end coordinate of the final big phase block, and "hp1" denotes the haplotype "1". "177969599:178064599_hp1" and "177869599:177969599_hp1" mean that this contig is concatenated from mini-contigs in small chunk (start coordinate: 177969599, end coordinate: 178064599, and haplotype: "1") and small chunk (start coordinate: 177869599, end coordinate: 177969599, and haplotype: "1").

Aquila outputs all raw contigs, even those <1 kb. For downstream analyses (e.g. to apply Merqury[24] or to discover variants), it is necessary to filter out small contigs.

**Combining multiple libraries of linked-reads into a single assembly**. Each library of 10X linked-read sequencing data uses the same set of barcodes, which makes the combination of multiple libraries at the stage of raw fastq reads difficult. To combine multiple libraries, Aquila reconstructs the original fragments with their corresponding reads of each library separately, after which barcodes are not required any more. All the DNA fragments from multiple libraries, and their corresponding reads, are then combined to perform haplotyping and all subsequent steps. The molecule haplotyping algorithm is efficient and makes haplotyping fragments for multiple libraries fast. The algorithm complexity is linear with the number of libraries, $\sim N_{libs}O(T_{onelib})$, where $N_{libs}$ = number of libraries and $T_{onelib}$ = time for finishing assembly of one library.

**Validation and evaluation methods for variations**. To validate the SNP calls from the assembled contigs, we used the GiaB benchmark call sets for both NA12878 and NA24385. They included 3,084,732 SNPs (10,210,585 homozygous and 1,874,147 heterozygous) for NA12878, and 3,076,552 SNPs (1,180,678 homozygous and 1,895,874 heterozygous) for NA24385. The benchmark allowed us to calculate both the sensitivity and the genotype accuracy of each SNP called from assemblies. We performed the same analysis on the small indel calls with the small indel callset 0.6 from GiaB (NA12878 ftp://ftp-trace.ncbi.nlm.nih.gov/giab/ftp/release/NA12878_HG001/latest/GRCh38/, NA24385 ftp://ftp trace.ncbi.nlm.nih.gov/giab/ftp/release/AshkenazimTrio/HG002_NA24385_son/latest/ GRCh38/). For SV validation we applied svviz2 (https://github.com/nspies/svviz2) with PacBio reads (NA12878: ftp://ftp-trace.ncbi.nlm.nih.gov/giab/ftp/data/NA12878/NA12878_PacBio_MtSinai/ and NA24385: ftp://ftp-trace.ncbi.nlm.nih.gov/giab/ftp/data/AshkenazimTrio/HG002_NA24385_son/PacBio_MtSinai_NIST/).

We classified SVs into three categories: Alu, Tandem Repeats, and Other. We selected SVs from 250 to 350 bp for classification as Alu elements by alignment to the AluY consensus sequence, and only SVs that matched at 80% or more were labeled as Alus. We used tandem repeats finder (trf[38]) to label repeats/non-repeats for all non-Alu-SVs, and the repeat percentage was calculated. Sequences that did not meet the Alu alignment criterion and did not contain a tandem repeat were labeled "Other".

For assemblies of different libraries, the different assemblies could identify different SV candidates. To analyze the overlapping and unique SV calls of three libraries of the same individual, we also merged each SV from different libraries. We applied the same merging criteria when we called homozygous/heterozygous SV from two haplotypes. If the coordinates of two deletions overlapped we defined them to be the same deletion, and if the breakpoints of two insertions were within 20 bp, we defined them to be same insertion.

**Ancestral analysis**. For each called SV, we extracted the left and right flanking 500 bp, and aligned them to the Orangutan and Chimpanzee references. For deletion calls, if the end coordinate of the left flanking sequence was within 2 bp of the start coordinate of the right flanking sequence, this SV was defined as "Ins_Ref" (insertion on reference—an actual insertion). If the distance between the end coordinate of the left flanking sequence and the start coordinate of the right flanking sequence was the approximate length of the called deletion (0.9*SV_size −1.1* SV_size), this SV was defined as "Del_Tar" (deletion on target—an actual deletion). For insertion calls, if the end coordinate of the left flanking sequence was within 2 bp of the start coordinate of the right flanking sequence, this SV was defined as "Ins_Tar" (insertion on target—an actual insertion). If the distance between the end coordinate of the left flanking sequence and the start coordinate of the right flanking sequence was the approximate length of the called deletion (0.9*SV_size−1.1* SV_size), this SV was defined as "Del_Ref" (deletion on reference—an actual deletion).

We performed multiple sequence alignments (for each SV and its aforementioned flanking regions annotated as "Ins_Ref", "Del_Tar", "Ins_Tar" and "Del_Ref") with Muscle[39].

**Comparison of genotype frequencies**. We asked whether segregation of the SV alleles behaves as expected. For each locus, we have two chromosomes each from the sequenced individuals and one from the reference genome, for a total of 5, and therefore 18 possible combinations of ancestral and derived alleles among them (Supplementary Fig. S7a). We cannot observe loci in which all chromosomes are identical, leaving 16 patterns that contain one to four derived alleles. Population genetic principles state that derived alleles have systematically lower allele frequencies than ancestral ones. Indeed, for both insertions and deletions, the SV loci that have one derived allele (and four ancestral ones) are much more common than those that have two or more (Supplementary Fig. S7b). In 12 of the segregation patterns, genotypes of NA12878 and NA24385 differ. These can be arranged in six pairs where the individuals' genotypes are equivalent, for example, one heterozygote and one ancestral homozygote, with the reference carrying a derived allele (rightmost green box in Supplementary Fig. S7a). Distinguishing insertions and deletions, this gives 12 classes (bottom of Fig. 3a), each of which has two SV counts

that should be very similar to each other assuming that the two individuals do not come from very different populations. Indeed, the correlation between these equivalence classes is very high ($R^2 = 0.92$, Supplementary Fig. S7c, d).

**Reporting summary**. Further information on research design is available in the Nature Research Reporting Summary linked to this article.

## Data availability

Source data are provided with this paper. The raw sequencing data can be downloaded from the Sequence Read Archive https://www.ncbi.nlm.nih.gov/bioproject/PRJNA527321/ and its BioProject accession number is PRJNA527321[18]. Assemblies and VCFs can be found at http://mendel.stanford.edu/supplementarydata/zhou_aquila_2021/. Source Data and Python scripts for Figs. 2 and 3 and Supplementary Fig. 6 are provided at the same link.

## Code availability

Aquila can be found at https://github.com/maiziex/Aquila [40] and https://doi.org/10.5281/zenodo.4312158. For easy installation, install through Bioconda by "conda install aquila". Version 1.0.0 was used to generate the results in this paper.

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

## Acknowledgements

This research was supported by the Joint Initiative for Metrology in Biology (JIMB; National Institute of Standards and Technology). We would like to thank Noah Spies, Justin Zook, and Marc Salit for informative discussions, and Ziwei Chen for help with a phasing example.

## Author contributions

X.Z., D.L.D., and A.S. designed the architecture of Aquila; X.Z. implemented Aquila; X.Z., L.Z., and Z.W. did the analysis; X.Z. and A.S. wrote the paper with input from L.Z., Z.W., and D.L.D.

## Competing interests

The authors declare no competing interests.
