## [Peer Review File · Nature Communications]

Reviewer #1 (Remarks to the Author):

The manuscript describes a computational method called Aquila for diploid personal genome assembly and comprehensive variant detection, using linked-read sequencing techniques. Given the unique advantage of linked read sequencing to generate synthetic long reads and generate phased variant calls, it yields much more information than conventional Illumina short-read sequencing with one extra library preparation step. Therefore, improved genome assembly and variant calling algorithms would be useful for users to take advantage of the linked read technology (note that 10X Genomics does provide their own set of software tools to analyze linked read sequencing data, including genome assembly, genome alignment, etc). The development of improved bioinformatics tools to handle these types of data sets would be important for the human genetics community to better understand genome structure and perform genetic diagnosis for patients.

My main comments are below to help improve the manuscript:

1. The main issue is that 10X Genomics already discontinued their product and no longer supports linked read sequencing, so while the methods proposed in the manuscript can be useful, it logically needs to be demonstrated in other linked read sequencing technologies. (Almost exactly the same thing happened with Illumina: they developed/acquired and then discontinued their own synthetic long-read sequencing technology a few years ago, so while lots of data is out there, nobody really needs software tools to analyze them nowadays.) These are quite a few additional linked read sequencing technologies developed over the past a few years, but perhaps the more widely known ones are BGI's Single Tube Long Fragment Read (stLFR) technology (published in Genome Research last year) or USO's TELL-Seq (under review but published in BioRxiv). They technically do not call themselves as "linked read sequencing", but they are essentially developed under similar ideas as 10X Genomics's linked read technology and can be largely compatible with analytical tools developed for 10X Genomics with some tweaks. I think to make the method in this paper useful for the readership, it cannot be solely tested on a technology that is already dead, as users will not be using this technology any more. Evaluating it on similar technology would be ideal.
2. Unless I completely misunderstood the methods, the Aquila tool is not really doing a genome assembly. It does require a "High Quality Reference Genome" as input. This is apparent in Figure 1B, but not illustrated in Figure 1A. I think it is fair to call it something like reference-guided diploid genome assembly, but it is not a genome assembly tool by itself, and it requires LongRanger (a tool developed by 10X Genomics) to align the reads to a high-quality reference genome as the initial step. I do not think it is a limitation of the approach and this is not a criticism against the method, but it is somewhat misleading in title and abstract and introduction, until I read through Results.
3. I have a problem understanding the procedure about downsampling of data to ~60x was performed for Supernova2 assemblies since that is its optimal coverage. Here you are generating six libraries, each having 100X coverage (including one with 192X coverage), and then compare the performance with Supernova using 60X coverage data. This is not a fair comparison. It would be ideal to use the exact same input data set of ~60X. (Additionally, I was not aware that supernova requires 60X coverage but not more coverage, please double check with 10X Genomics's technical support about this since it does not quite make sense to me though I am not completely sure if this is indeed the case; but again since 10X genomics no longer supports linked read sequencing maybe they no longer support supernova either).

4. Continued on the above issue, did not Supernov perform de novo whole-genome assembly? Then why would you compare the contig N50 between Aquila with supernova? Yet it looks like supernova generates better results than Aquila, which is more confusing to me as it is somewhat counter-intuitive. In any case, I believe that what you should emphasize is “fraction of genome in a diploid state” in their assembly, and simply acknowledge that the assembly completeness is not as good as supernova, since the main purpose is to generate diploid assembly rather than more continuous assembly.

5. I have difficulty to understand the unique aspect of the proposed algorithm Aquila, compared to a conventional approach such as Supernova, when doing genome assembly. In both Figure 1 and in the “Aquila architecture and workflow” subsection of the Results section, the authors can consider adding more details to describe the novel or unique aspects of Aquila that differentiate it from existing state of the art. For example, the main goal seems to be generating high fraction of genome in a diploid state, using reference-guided assembly, rather than just a more complete genome assembly (which is what PacBio or Nanopore is better at producing, given the very small N50 values shown in Table 1).

Some additional minor issues are listed below:

1. The sentence “Assemblies cover >95% of the human reference genome, with over 98% in a diploid state” is out of context in Abstract. It seems that a few sentences before it was deleted, which describes what data sets are used.

2. The statement “the drawback of both approaches is that they exhibit poor base-pair level accuracy” on PacBio and Nanopore is no longer relevant, since both methods can now generate very high base-level accuracy while maintaining much longer read length than Illumina sequencing, through different library preparation techniques. I do not think it is a valid argument here to criticize these competing technologies. (I know that they did mention CCS reads in the same paragraph, but this concept is already replaced by HiFi reads, so the CCS concept no longer exists). The main limitation is their per-base cost, compared to Illumina sequencing, as well as the stringent requirement of large quantities of input DNA materials (due to the nature of single-molecule sequencing) compared to linked-read sequencing.

3, For the statement “We tested its performance with six libraries of 10x linked-reads data for NA12878 and NA24385” in Introduction, it would be good to add a citation (if available, since it seems that the data is actually generated in the authors themselves) and also add some description of the data set (such as coverage, etc). The simple reason is that too many groups have generated data sets on the same subject, including the 10X Genomic company itself and also genome in a bottle consortium. The quality of the various data sets can vary greatly. It is important to give a little more details when mentioning data sets that are used in the manuscript.

4. Continuing the previous comments, since they used GIAB throughout the manuscript for comparison, I do not understand why they do not use GIAB’s linked read data. Maybe the data quality is not as good as what the authors have generated, or maybe there are other reasons. But at a bare minimum, the existing (and widely used) public data sets should be used for testing the methods, since many other papers already used the same data sets to produce results that can be directly compared to the current manuscript.

5. Although it is not a major reason to use linked read sequencing, SNP and indel calling accuracy is an important factor to consider. From what I read, Aquila actually performs poorly compared to FreeBayes and LongRanger (in Table 2, for both HG001 and HG002). Note that I am not sure why L5+L6 results are not shown for competing approaches. The indel performance looks better (in Table 3, for both HG001 and HG002), which is somewhat strange since indels are not influenced by assembly quality (again, L5+L6 results are not shown). Additionally, I do not understand why false positive rates for Aquila is low, given that it produced twice as many calls as the gold standard GIAB; the information is not consistent (i.e. false positive is around ~50K yet true positive is around ~500K).

6. I also think given that there are 1,007,313 in L3 set vs GiaB's 531,382 (i.e. basically a two fold difference), the presentation of results is a bit misleading. I am not saying that the extra 500K indels are false positives, in fact, many of them may well be true positives that are not found by GIAB, though the chance is not very high. Again, SNP or indel calling is not something that linked read sequencing has an intrinsic advantage compared to conventional sequencing, and I do not see why focusing on this aspect in the paper using so many Tables and so much texts in the paper. The main advantage that the paper should present is the ability to generate phased (and diploid) genome assemblies, as well as phased variant calls which are not really stressed in any of the figures or tables (except table 5).

7. Table 1: L1_supernova is not shown in the Table, it may be accidentally missed by the authors. It is also important to mention in Table legend about how supernova is used (i.e. whether 60x down sampling is used), since the Table lists the exact sequencing coverage of each of the data set.

8. I feel that they can consider adding more informative display items. This is a paper about a method for diploid genome assembly and phased variant detection. Right now there are 3 figures, figure 1 is just an illustration, figure 2 is a comparison of variant calls with no relevance to either genome assembly or phased variant calls. I am not sure what figure 3 is trying to show: for example the "Repeat" track is almost completely green (with a tiny bit of yellow), yet repeats detected by TRF should be very frequently occurring in genome; the "Contigs" track is confusing, I cannot really tell what "average number of contigs per base pair" really means. I cannot really tell what this figure is trying to present to readers. I think the authors can do a much better job making informative figures illustrating the performance of the method, say for example, demonstrating how it can detect two non-synonymous SNPs in the same phase, which cannot be done by conventional Illumina short-read sequencing.

Reviewer #2 (Remarks to the Author):

The authors presented a reference guided phasing method using 10X Genomics linked reads for more accurate SNPs, short indels, and (relatively short) SV detection. Aquila produces comparable or better results compared to Supernova, which was the only assembler available for 10X Genomics linked-reads. However, a few major concerns need to be addressed to make the manuscript more compelling. I have reviewed this manuscript for three times, but am really disappointed with the little improvements I see.

Major concerns

1. What is unique or novel to Aquila's haplotyping module compared to HapCUT2 (Edge et al,

Genome Res. 2017) or WhatsHap (Patterson et al, J Comp. Biol. 2015)? Direct comparison to the num. of phased / total variants, phase block N50, switch error statistic summary will be useful.

2. It is difficult to agree that Aquila is producing overall better results than Supernova. Contig N50s seem similar or lower. Fraction of reference genome covered is not always representing the completeness of capturing heterozygous variants. There are more missing sequences in the Aquila assembly, especially in the haplotype specific regions when comparing to NA24385 Supernova 2.0 assembly (KAT spectra-cn plots from Mapleson et al. Bioinformatics, 2016 shows largely missing heterozygous portion in Aquila assemblies but not in supernova assemblies). Reporting the k-mer completeness as done in Merqury (Rhie et al. bioRxiv, 2020) along with NG50, NGA50s will show the true sensitivity of capturing genetic variations.

3. The base level QV is not mentioned in the manuscript. However, measuring QV from 21-mers on the L5+L6 assembly using Merqury was surprisingly lower than the Supernova assembly (Q38 vs. Q50). I am confused, I had never seen an Illumina contig going below Q40.

4. NA24385 is a male. How is Aquila handling sex chromosomes?

5. Phased blocks seem to be large inferred from Fig. 3. However, no summary statistics are provided as a table. Adding phase block N50s along with number of phased / total variants to Table 1 will be useful to understand the output better. In addition, supplemental table providing above stats on each haplotyping stage will be helpful to understand the process.

6. Is it possible to increase contig continuity? Ideally, one would expect to have two contigs (haplotigs) per phase block. Why are the contigs overly fragmented in each phased block?

7. Related to 5., how are the reads with no variants handled? One possible way would be generating a pseudo haplotype assembly using one haplotype (ie. hp1) and the un-binned reads, and provide the other haplotype (ie. hp2) as alternate haplotigs.

8. Compare the overall contigs from each haplotype in MHC region and HLA typing results in NA12878 (See Supp. Table 9 from Dilthey et al., Nat Genet. 2015).

9. On page 5, the authors mention "greater than 99.5% of anybody's two haplomes is identical to the reference". This needs a reference, or be removed. This is also counter intuitive to the fact that Aquila is only covering 95% of the reference. If the haplomes are 99.5% identical, why is Aquila only covering only 95% of the reference?

10. Fig. 2f is very confusing. Why not just show the status of the Derived alleles in Fig. F? Including the ancestral variants makes the figure very complicated. Also, does not help the argument made for Alu insertions as derived alleles. Does the pattern in Fig. G remain when removing the ancestral allele calls?

We deeply appreciate the thoughtful comments by the reviewers, which we address point by point below. Before we get into the details we would like to take the opportunity to make some overarching comments to explain both the motivation and the history of the development of Aquila. It would be difficult to do this as part of the paper, but for the review process it provides some important information that the reviewers might consider.

When the linked read idea first appeared 'on the scene' <10 years ago it showed significant conceptual promise but its actual usefulness for comprehensive human variation detection soon turned out to be limited because, after all, it is still a short-read approach. We hypothesized, however, that improvements would be possible if libraries were generated from optimally isolated long DNA fragments; and if synergistic software was written that really leveraged the strengths of the "linking" of the short reads. As a result of multiple consultations with David Jaffe and 10x, we generated several libraries from our own long-fragment DNA preps and evaluated the performance of Supernova in both simulations and with the real data, as a function of different library construction and sequencing parameters, analyzed on both the original Supernova and Supernova2. This led to two papers: (1) Zhang et al, De novo diploid genome assembly for genome-wide structural variant detection. *NAR Genomics and Bioinformatics*. Mar 2020; (2) Zhang et al, Assessment of human diploid genome assembly with 10x Linked-Reads data. *Gigascience*. Nov 2019.

Supernova2's performance was impressive and our results were similar to those presented in David's GR paper. However, we uncovered shortcomings that were glossed over in his paper, the most important of which was the fact that a large fraction of the human genome (>20% in our analyses) was not in a diploid state. This explained the relatively poor performance of Supernova-based SNP detection, at ~91% sensitivity as measured against the GiAB benchmark (Zhang et al, 2020); however, Supernova-based assemblies had fairly good performance for insertions and deletions (Zhang et al, 2020).

It makes complete sense to use Supernova as an inexpensive assembly solution for species without a reference; what else can one do? It has to be a true de novo assembly. However, for genomes that do have a high quality reference, why not make use of its high information content to guide an assembly? Thus, Aquila is the implementation of the concept that long-range information from the reference can be leveraged to help construct a personal genome: that one virtually arranges the original long fragments of a DNA prep along the reference; that they can be partitioned into parent 1 and parent 2 on the basis of heterozygous SNPs; and that reads in local bins can then be de-novo assembled free of variation (as they originate from only one parent at a time), affording optimal conditions for an assembler which now does not need to deal with variation-induced ambiguity.

Aquila's assemblies are just a means to an end: to identify the vast majority of variants (SNPs, small indels, and longer indels), in a personal genome, with a single data type (linked reads), and a single computational approach (Aquila), at low cost (ILMN sequencing). The assembly statistics, while they are informative in comparisons between libraries and different approaches, are not the end goal. The end goal is to produce a single cost-effective solution for discovering and phasing a personal genome's vast majority of variants at high accuracy.

Therefore, evaluation of Aquila needs to focus on the use case for which it was designed: variant discovery in a personal genome in a species for which significant resources such as a reference as well as variation benchmarks exist. It does not make sense to confine evaluation of Aquila to the metrics that are used to evaluate de novo assemblers or haplotyping tools. What matters is the variant discovery and the cost at which it can be achieved; we present assembly metrics as important auxiliary information but ask that they are not considered the focus of Aquila's evaluation. Evaluation of Aquila needs to rest on (1) its performance for comprehensive variant discovery (SNPs, small indels, larger indels), which we show to be high, given (2) the cost of generating the data, which is reasonably low, and (3) the complexities of the overall approach, in (3a) lab and (3b) silico, which are also low.

Reviewer #1 (Remarks to the Author):

The manuscript describes a computational method called Aquila for diploid personal genome assembly and comprehensive variant detection, using linked-read sequencing techniques. Given the unique advantage of linked read sequencing to generate synthetic long reads and generate phased variant calls, it yields much more information than conventional Illumina short-read sequencing with one extra library preparation step. Therefore, improved genome assembly and variant calling algorithms would be useful for users to take advantage of the linked read technology (note that 10X Genomics does provide their own set of software tools to analyze linked read sequencing data, including genome assembly, genome alignment, etc.). The development of improved bioinformatics tools to handle these types of data sets would be important for the human genetics community to better understand genome structure and perform genetic diagnosis for patients.

We sincerely appreciate the reviewer's succinct summary of the motivation and significance of our contribution.

My main comments are below to help improve the manuscript:

1. The main issue is that 10X Genomics already discontinued their product and no longer supports linked read sequencing, so while the methods proposed in the manuscript can be useful, it logically needs to be demonstrated in other linked read sequencing technologies. (Almost exactly the same thing happened with Illumina: they developed/acquired and then discontinued their own synthetic long-read sequencing technology a few years ago, so while lots of data is out there, nobody really needs software tools to analyze them nowadays.) These are quite a few additional linked read sequencing technologies developed over the past a few years, but perhaps the more widely known ones are BGI's Single Tube Long Fragment Read (stLFR) technology (published in Genome Research last year) or USO's TELL-Seq (under review but published in bioRxiv). They technically do not call themselves as "linked read sequencing", but they are essentially developed under similar ideas as 10X Genomics's linked read technology and can be largely compatible with analytical tools developed for 10X Genomics with some tweaks. I think to make the method in this paper useful for the readership, it cannot be solely tested on a technology that is already dead, as users will not be using this technology any more. Evaluating it on similar technology would be ideal.

We were very sad (and argued with them) that 10x decided to discontinue the technology that gave birth to the company, while we had finished the first version of our paper and finalized Aquila. Fortunately, stLFR and Tell-Seq appeared as competing approaches and are still available. Tell-Seq works with Aquila out of the box; for stLFR we implemented a custom version of Aquila (Aquila_stLFR) because of the much larger number of barcodes. We ran Aquila_stLFR on the available BGI data, but because the libraries were not made with the same care as our 10x libraries the results are not as good (see below). At any rate, Aquila_stLFR is available on GitHub, has been used extensively by the BGI folks and others, and has been downloaded more than 470 times so far. We believe that Aquila will accelerate the uptake of the new linked-reads technologies, and vice versa, in a virtuous cycle once Aquila is published.

NA24385_GIAB_stLFR (hg19)	TP	FP	FN	recall	precision
DEL (>= 50bp)	3277	8925	922	78.0%	26.9%
INS (>= 50bp)	1101	216	4341	20.2%	83.6%

2. Unless I completely misunderstood the methods, the Aquila tool is not really doing a genome assembly. It does require a "High Quality Reference Genome" as input. This is apparent in Figure 1B, but not illustrated in Figure 1A. I think it is fair to call it something like reference-guided diploid genome assembly, but it is not a genome assembly

tool by itself, and it requires LongRanger (a tool developed by 10X Genomics) to align the reads to a high-quality reference genome as the initial step. I do not think it is a limitation of the approach and this is not a criticism against the method, but it is somewhat misleading in title and abstract and introduction, until I read through Results.

We thank the reviewer for making this important point, which seems to lead to confusion. It was not our intention to pretend that Aquila is a de novo assembly method, although it uses local de novo assembly of reads. We changed Figure 1A to indicate the reference genome, as in 1B. We also changed the title and abstract to clarify that Aquila makes use of a reference sequence.

3. I have a problem understanding the procedure about downsampling of data to ~60x was performed for Supernova2 assemblies since that is its optimal coverage. Here you are generating six libraries, each having 100X coverage (including one with 192X coverage), and then compare the performance with Supernova using 60X coverage data. This is not a fair comparison. It would be ideal to use the exact same input data set of ~60X. (Additionally, I was not aware that supernova requires 60X coverage but not more coverage, please double check with 10X Genomics's technical support about this since it does not quite make sense to me though I am not completely sure if this is indeed the case; but again since 10X genomics no longer supports linked read sequencing maybe they no longer support supernova either).

We have clarified in the text (page 5) that these libraries had been built for our two Supernova papers (cited above in the preamble). Supernova has a 1.2B read limit, corresponding to approximately 60x of a human genome. In our Gigascience paper on Supernova performance, we noticed that the "down-sampling" mode in Supernova to 60X is necessary; upon inquiry Jaffe told us that he doesn't believe that more coverage is necessary, so he set this as a hard limit to avoid overtaxing compute resources. We have since applied Aquila to lower-coverage genomes with worse DNA preparations, and it works fine, though of course not as well as with deeper coverage. (If people didn't take care to prepare good DNA for ONT and sequence at reasonable coverage, ONT would not perform as well either, for example.)

The point of our paper is not to compare Aquila to Supernova, as they have nonoverlapping use cases: Aquila for a species/individual with a reference, Supernova for species without a reference. But we have to show a comparison, people ask for it. With respect to coverage, we cannot allow Supernova's limitation to unfairly penalize our approach, which we show does not have a coverage limitation and in fact improves with increasing coverage (e.g., see the L5+L6 combined library statistics compared to L5 and L6 separately).

4. Continued on the above issue, did not Supernova perform de novo whole-genome assembly? Then why would you compare the contig N50 between Aquila with supernova? Yet it looks like supernova generates better results than Aquila, which is more confusing to me as it is somewhat counter-intuitive. In any case, I believe that what you should emphasize is "fraction of genome in a diploid state" in their assembly, and simply acknowledge that the assembly completeness is not as good as supernova, since the main purpose is to generate diploid assembly rather than more continuous assembly.

It is possible to achieve higher contiguity but only at the expense of diploid fraction. For example, Supernova gets the highest N50 for library L2, but does so by achieving a woeful 58.9% diploid fraction, which results in the L2 assembly providing the worst detection of heterozygous variants. For libraries with high fragment coverage (L3, L5, and L6), Aquila has better N50 and genome coverage than Supernova.

At any rate, the reviewer correctly points out that the fraction of genome in a diploid state is where Aquila really outperforms Supernova. This cements the initial motivation for developing Aquila (see preamble above), which we now state explicitly in the beginning of the Results section as well (see next point).

5. I have difficulty to understand the unique aspect of the proposed algorithm Aquila, compared to a conventional approach such as Supernova, when doing genome assembly. In both Figure 1 and in the "Aquila architecture and workflow" subsection of the Results section, the authors can consider adding more details to describe the novel or

unique aspects of Aquila that differentiate it from existing state of the art. For example, the main goal seems to be generating high fraction of genome in a diploid state, using reference-guided assembly, rather than just a more complete genome assembly (which is what PacBio or Nanopore is better at producing, given the very small N50 values shown in Table 1).

We hear the reviewer: that we're insufficiently clear about the combination of conceptual motivation behind Aquila, and how it implements a solution to that motivation. We understand the challenge in explaining our method, which combines both of the two traditional approaches: read mapping and de novo assembly. Aquila uses read mapping, but its end product is an assembly for variation discovery. To be extra clear about it we added this paragraph to the text at the beginning of the results section:

"The motivation of Aquila is to generate sufficiently long contigs from each parental haplotype such that variation can be discovered on the basis of pairwise alignment of these contigs to the reference. Contigs on the order of 100kb (implying ca. 60,000 contig breaks in a diploid 3Gb genome) are in principle sufficient to discover most variation as long as the vast majority of the genome is recovered in a diploid state. Hence, the contiguity of the assembly is less important than the diploid nature of it. Aquila's use case is distinct from that of de novo assembly, where the luxuries of a high quality reference sequence and extensive variation benchmarking resources do not exist. Aquila's motivation is to leverage these luxuries in the service of comprehensive variation discovery."

Some additional minor issues are listed below:

1. The sentence "Assemblies cover >95% of the human reference genome, with over 98% in a diploid state" is out of context in Abstract. It seems that a few sentences before it was deleted, which describes what data sets are used.

We have clarified this in the abstract.

2. The statement "the drawback of both approaches is that they exhibit poor base-pair level accuracy" on PacBio and Nanopore is no longer relevant, since both methods can now generate very high base-level accuracy while maintaining much longer read length than Illumina sequencing, through different library preparation techniques. I do not think it is a valid argument here to criticize these competing technologies. (I know that they did mention CCS reads in the same paragraph, but this concept is already replaced by HiFi reads, so the CCS concept no longer exists). The main limitation is their per-base cost, compared to Illumina sequencing, as well as the stringent requirement of large quantities of input DNA materials (due to the nature of single-molecule sequencing) compared to linked-read sequencing.

We agree with the reviewer and have toned down this paragraph.

- 3, For the statement "We tested its performance with six libraries of 10x linked-reads data for NA12878 and NA24385" in Introduction, it would be good to add a citation (if available, since it seems that the data is actually generated in the authors themselves) and also add some description of the data set (such as coverage, etc.). The simple reason is that too many groups have generated data sets on the same subject, including the 10X Genomic company itself and also genome in a bottle consortium. The quality of the various data sets can vary greatly. It is important to give a little more details when mentioning data sets that are used in the manuscript.

We clarified the origin of the sequence data in the last paragraph of the Introduction.

4. Continuing the previous comments, since they used GIAB throughout the manuscript for comparison, I do not understand why they do not use GIAB's linked read data. Maybe the data quality is not as good as what the authors have generated, or maybe there are other reasons. But at a bare minimum, the existing (and widely used) public data sets should be used for testing the methods, since many other papers already used the same data sets to produce results that can be directly compared to the current manuscript.

We respectfully disagree with the reviewer on this one (though L7 in Table 1 is actually 10x's data from GIAB). It is important to generate high quality libraries, and the public data does not allow us to highlight the strengths of our approach. For example, one would not subject ONT performance to evaluation using suboptimal DNA preps. That said, we have a manuscript in preparation that uses all available 10x data (34 individuals from 5 populations, including all of the public sets), through Aquila assemblies and a Multiple Alignment-based Refinement of SVs (<https://github.com/maiziex/MARS>), and we still get excellent SV detection, which is the focus of that manuscript.

5. Although it is not a major reason to use linked read sequencing, SNP and indel calling accuracy is an important factor to consider. From what I read, Aquila actually performs poorly compared to FreeBayes and LongRanger (in Table 2, for both HG001 and HG002). Note that I am not sure why L5+L6 results are not shown for competing approaches.

Aquila does both mapping-based and assembly-based SNP discovery, and they are combined to get the best of both worlds. We note this in the Results section: "SNPs are also detected but this assembly-based set is merged with the initial set of SNPs identified on the basis of the barcode-aware read alignment performed by the Haplotyping module." The "Aquila" rows in Table 2 present only assembly-based SNP detection to highlight the surprisingly good performance of assembly-only based SNP calling. We added some clarifying edits. The point of showing L5+L6 is to show improvement for Aquila when libraries are combined, as noted in the text.

The indel performance looks better (in Table 3, for both HG001 and HG002), which is somewhat strange since indels are not influenced by assembly quality (again, L5+L6 results are not shown). Additionally, I do not understand why false positive rates for Aquila is low, given that it produced twice as many calls as the gold standard GIAB; the information is not consistent (i.e. false positive is around ~50K yet true positive is around ~500K).

The numbers we cite for Aquila is the totality of called indels; the performance evaluation vs GIAB is only performed in the GIAB high-confidence regions (which are provided by GIAB in a bed file).

6. I also think given that there are 1,007,313 in L3 set vs GiaB's 531,382 (i.e. basically a two-fold difference), the presentation of results is a bit misleading. I am not saying that the extra 500K indels are false positives, in fact, many of them may well be true positives that are not found by GIAB, though the chance is not very high. Again, SNP or indel calling is not something that linked read sequencing has an intrinsic advantage compared to conventional sequencing, and I do not see why focusing on this aspect in the paper using so many Tables and so much texts in the paper. The main advantage that the paper should present is the ability to generate phased (and diploid) genome assemblies, as well as phased variant calls which are not really stressed in any of the figures or tables (except table 5).

The notion that linked read sequencing is not for SNPs and small indels is based on previous analytical approaches that did not leverage the strengths of the data (as well as suboptimal library generation). Aquila changes this. The whole point of an assembly is to discover variants. We show definitively that Aquila performs extremely well in comprehensive variant detection for a personal genome.

7. Table 1: L1_supernova is not shown in the Table, it may be accidentally missed by the authors. It is also important to mention in Table legend about how supernova is used (i.e. whether 60x down sampling is used), since the Table lists the exact sequencing coverage of each of the data set.

We have fixed this.

8. I feel that they can consider adding more informative display items. This is a paper about a method for diploid genome assembly and phased variant detection. Right now there are 3 figures, figure 1 is just an illustration, figure 2 is a comparison of variant calls with no relevance to either genome assembly or phased variant calls. I am not sure what figure 3 is trying to show: for example the "Repeat" track is almost completely green (with a tiny bit of yellow), yet repeats detected by TRF should be very frequently occurring in genome; the "Contigs" track is

confusing, I cannot really tell what “average number of contigs per base pair” really means. I cannot really tell what this figure is trying to present to readers. I think the authors can do a much better job making informative figures illustrating the performance of the method, say for example, demonstrating how it can detect two non-synonymous SNPs in the same phase, which cannot be done by conventional Illumina short-read sequencing.

The meat of the results is in the Tables, but the figures are important too: Figure 1 explains the methodology, which is essential. Figure 2 illustrates the variant calls, as the reviewer correctly notes; variant calls are the point of Aquila, so this is a highly relevant figure, especially the validation rates. Figure 3 is there to give readers comfort that our variant calls are not weirdly distributed across the genome in some way. Aquila is new, linked reads are not well known, so we need to show some expected behavior. That said, we removed the TRF track. With respect to showing that we do phasing well, the last track of Figure 3A as well as Tables 5 and Supplementary Table 7 should prove to anyone that we phase nonsynonymous SNPs even if they are very far apart.

Reviewer #2 (Remarks to the Author):

The authors presented a reference guided phasing method using 10X Genomics linked reads for more accurate SNPs, short indels, and (relatively short) SV detection. Aquila produces comparable or better results compared to Supernova, which was the only assembler available for 10X Genomics linked-reads. However, a few major concerns need to be addressed to make the manuscript more compelling. I have reviewed this manuscript for three times, but am really disappointed with the little improvements I see.

We appreciate the reviewer’s careful reading to our paper – and the opportunity to respond to some of the criticism raised, in this and previous reviews.

Major concerns

1. What is unique or novel to Aquila’s haplotyping module compared to HapCUT2 (Edge et al, Genome Res. 2017) or WhatsHap (Patterson et al, J Comp. Biol. 2015)? Direct comparison to the num. of phased / total variants, phase block N50, switch error statistic summary will be useful.

This is a very good question that we did not adequately explain in the initial submission. As described in the preamble, Aquila is a diploid assembly-based WGS variant detection pipeline. Phasing is not Aquila’s main aim. We designed our own phasing module, which selects high quality SNPs to help partition linked reads to one or the other parent so that we can perform local assembly that is free of variation. Long phase blocks or maximizing total number of phased variants in our phasing module are not goals in and of itself, and the assembly module in Aquila actually needs to cut large phase blocks into small chunks for local assembly. In Aquila, probabilistic phasing occurs only in the beginning of the pipeline; later phasing is deterministic because, by virtue of the assembly, the alleles of non-SNP variants and previously unphased SNPs are linked to already-phased SNPs via the contig sequence.

We did generate phasing statistics with HapCUT2 (Table 5). HapCUT2 failed to run on four libraries.

Finally, in further response to this point, we have implemented an improvement in the haplotyping module. Aquila now can also use common SNPs from 1000Genomes to help partition linked reads, instead of individual variant calls.

2. It is difficult to agree that Aquila is producing overall better results than Supernova. Contig N50s seem similar or lower. Fraction of reference genome covered is not always representing the completeness of capturing heterozygous variants. There are more missing sequences in the Aquila assembly, especially in the haplotype specific regions when comparing to NA24385 Supernova 2.0 assembly (KAT spectra-cn plots from Mapleson et al. Bioinformatics, 2016 shows largely missing heterozygous portion in Aquila assemblies but not in supernova assemblies). Reporting the k-mer completeness as done in Merqury (Rhie et al. bioRxiv, 2020) along with NG50, NGA50s will show the true sensitivity of capturing genetic variations.

We understand that k-mer statistics are important for de novo assemblies of organisms that do not have a reference or variant benchmarks. However, as we have stated in the preamble above and now clarified further in the text, when luxury resources (reference, benchmarks) are available, other metrics of accuracy such as k-mer statistics are secondary. In every single one of our supernova assemblies, >20% of the reference is covered by only one contig; by contrast, *using the exact same sequence data*, Aquila misses 5%. Similarly, Supernova has ca. 91% accuracy in GiaB benchmark SNP detection, Aquila 99%. These results are unequivocal. Aquila performs better than Supernova. Nonetheless, we ran Merqury on L3 (NA12878) for both Aquila and Supernova assemblies. Aquila got 98.7624 and Supernova got 98.6877 completeness. See plots below.

Aquila_L3 (below) all 2266174202 2294572208 98.7624

Supernova_L3 (below) all 2264461577 2294572208 98.6877

3. The base level QV is not mentioned in the manuscript. However, measuring QV from 21-mers on the L5+L6 assembly using Merqury was surprisingly lower than the Supernova assembly (Q38 vs. Q50). I am confused, I had never seen an Illumina contig going below Q40.

Thank you for raising this issue. QV is below 40 when small contigs (<= 1kb) are included. Because they represent mostly noise in the form of repeat collapses, they should be filtered out before running Merqury. We now clarify this in the Supplementary Information (Aquila output) and GitHub. After filtering, the QV is 45. We also measured

the QV from HiFi reads and it was 39. NB: The SNP/Indels/SV calling from the assembly of this HiFi library is often used as a validation dataset.

4. NA24385 is a male. How is Aquila handling sex chromosomes?

Thank you for this question, which led us to clarify how Aquila handles sex chromosomes in the GitHub repository. Aquila cannot handle male-specific portions of the Y chromosome. For X chromosomes in females, Aquila processes them as autosomes and uses 23 to label them. For X chromosomes in males, if Aquila can partition linked-reads into two haplotypes, such as in the pseudoautosomal region, it can perform a diploid assembly, otherwise, Aquila performs a haploid assembly.

5. Phased blocks seem to be large inferred from Fig. 3. However, no summary statistics are provided as a table. Adding phase block N50s along with number of phased / total variants to Table 1 will be useful to understand the output better. In addition, supplemental table providing above stats on each haplotyping stage will be helpful to understand the process.

The reviewer may have missed Table 5, which provides the requested summary statistics (PB N50, long switch errors, mismatch, # phased variants in GIAB by evaluating with GIAB gold standard) for our phasing module. We only have one stage for the phasing module.

6. Is it possible to increase contig continuity? Ideally, one would expect to have two contigs (haplotigs) per phase block. Why are the contigs overly fragmented in each phased block?

This is an important point. Short read assemblies tend to break at simple sequence repeats that exceed the assembler's maximum k-mer size, and at other repeats with longer repeat units. Such repeats are fairly common in the human genome. We analyzed the nature of the contig breaks and most of them are indeed located in repeats.

7. Related to 5., how are the reads with no variants handled? One possible way would be generating a pseudo haplotype assembly using one haplotype (i.e. hp1) and the un-binned reads, and provide the other haplotype (i.e. hp2) as alternate haplotigs.

Reads from the same original DNA fragment are linked together by the fragment's barcode. Thus, we only need one read of a fragment to fall on a SNP heterozygosity in order to sort all reads of the fragment into the parent 1 or the parent 2 bin. As long as a read that does not cover a variant is linked to a read that does cover a heterozygosity, it will be assigned to the correct parental bin.

8. Compare the overall contigs from each haplotype in MHC region and HLA typing results in NA12878 (See Supp. Table 9 from Dilthey et al., Nat Genet. 2015).

This is a great idea. We performed this analysis and have added Supplementary Tables 8 and 9 to show Aquila's performance in calling MHC alleles, and added a note regarding this at the end of the results.

We also note that the latest MHC benchmark for HG002 (<https://github.com/NCBI-Hackathons/TheHumanPangenome/tree/master/MHC/>) which is currently in press at Nature Communications is generated from local assemblies. The overall algorithm is similar to Aquila: to partitioning of reads (linked-reads and long reads) into two haplotypes and local assembly in MHC regions. It covers 94% of the MHC and 22368 variants smaller than 50 bp, 49% more variants than a mapping-based benchmark. An Aquila assembly from 10x linked-reads was incorporated in this paper to generate the benchmark.

9. On page 5, the authors mention "greater than 99.5% of anybody's two haplomes is identical to the reference". This needs a reference, or be removed. This is also counter intuitive to the fact that Aquila is only

covering 95% of the reference. If the haplomes are 99.5% identical, why is Aquila only covering only 95% of the reference?

We now clarify that this was meant as a statement of biological fact: outside of telomeres and centromeres, humans have ca. 0.1% SNP heterozygosity, and adding in indels and SVs it might go up to ca 0.5% in total number of bases. Therefore, approximately 99.5% of each of our haplomes is identical to the reference, depending of course on geographic origin. Aquila covers 95% of the reference because it cannot cover long repeats (such as centromeres and telomeres). We have added 'outside of telomeres and centromeres' to the text to clarify.

10. Fig. 2f is very confusing. Why not just show the status of the Derived alleles in Fig. F? Including the ancestral variants makes the figure very complicated. Also, does not help the argument made for Alu insertions as derived alleles. Does the pattern in Fig. G remain when removing the ancestral allele calls?

Thank you for this suggestion, which prompted us to provide further clarification in the text. Figure 2F explains the logic behind setting the ancestral state and inferring which allele is derived. When the ape sequence matches the reference, the alternate allele is derived; when the ape sequence matches the alternate allele, the reference allele is derived. Without the alignment to ape and without showing the ape there is no such inference. It is key to relate the original call, which is based on the comparison to the reference sequence (insertion, deletion), to the ape alignment. Removing the ancestral allele call is not possible, as this analysis requires the ancestral allele call.

Reviewer #1 (Remarks to the Author):

I thank the authors to make an effort and address my previous comments. I shared the feeling of sadness that 10X Genomics decided to discontinue linked-read sequencing before its full potential is really revealed and recognized by the community.

Some additional comments below:

One suggestion that I made previously is that they should really stress the advantage of the novel computational approach. It is not just for finding variants, but for finding phased variants, given that such information is quite useful in many instances such as diagnosis of genetic diseases that are recessively inherited, and for finding mutations that interact with each other in cis (either physically by being in the same protein, or influenced by the same regulatory mechanism). Take this as an optional comment: use an example of a segment and show how two coding heterozygous variants are called in the same haplotype (you may even use IGV to show that half the linked-reads carry one allele that has two mutations in the same read). This ad hoc example may be more informative to tell users what are the main advantage of the new methods.

Please make sure that this is done “We will be submitting raw sequence data and assemblies to NCBI's SRA and Assembly databases”. The assembly is especially important and should not be omitted.

They are unable to address the minor comments #4, but they mentioned that they have another manuscript in preparation and claimed that the public data sets do not have as high quality as the data that were generated in the current study. I think this is fine, and I hope that they can provide the data sets publicly available if the quality and quantity is much higher than previous studies.

Reviewer #2 (Remarks to the Author):

The authors have addressed most of my initial concerns over the revision. Pleased to see benchmark comparison against HapCUT2, as well as the MHC phasing results. Adding “reference-assisted” in the title much better communicates the identity of Aquila.

Below are questions / comments on incompletely addressed concerns in the revised manuscript or response.

1. Table 5: format numbers consistently with commas. Apologies for overlooking this table in my prior review.
2. The supplementary note now states that contigs shorter than 1kbp needs to be removed prior to evaluate the base QV. However, this means variants called from these shorter contigs be less reliable. Are these contigs also removed prior to variant calling?
3. The spectra-cn plot of the L3 assembly (NA12878) the authors attached shows good completeness overall. However, lots of false duplications (3, 4, >4x on the ~45x multiplicity), which is expected from overlapping mini contig boundaries. This is one of the common assembly errors misleading to false duplication calls if not properly handled. How are these handled? A brief discussion, along with the <1kbp contigs from 2., should be addressed in Methods or Discussion.

4. Although the spectra-cn plot of NA12878 seems more complete, the NA24385 L5+L6 still suffers from missing heterozygous sequences (completeness measured with Merqury was 94.2% for hap1 and 93.2% for hap2, combined hap1 + hap2 96.2%). Is this because the Chr. Y was not handled in the pipeline? Or the genome being more diverged from the reference? Are the variants on Chr. Y completely discarded, thus not creating any contigs? Stating in github page is not enough. As this is a current limitation of Aquila, it should be stated in the Results or Discussions. Likewise, if the Chr. Y was excluded for measuring Genome Fraction (%) in Table 1, clarify in the legend. Below is the spectra-cn plot and the spectra-asm plot of the combined L5+L6 contigs. The black hump at ~30x is showing the missing portion. (See attached file)

5. Related to 4., the authors claim k-mer based completeness metrics are only valid when reference genomes are not available, however this underrepresents the advantage of the local de novo assembly of Aquila. Showing k-mer based metrics will complementary provide a more complete picture of the solidity that Aquila delivers as it provides an absolute measure of the genome coverage, lifting the biases caused by the reference.

6. I have asked on page 5, “greater than 99.5% of anybody’s two haplomes is identical to the reference” be referenced, or removed. The authors clarified in the revised text that they measure outside of the telomeres and centromeres, and responded their source of inference of the 99.5% was based on SNPs, indel, and SVs, presumably from the two human genomes used in this study genotyped with Aquila. If there is no formal reference, it has to be addressed how this number was obtained in the main text.

7. Page 13 line 312-314: Please use “individual’s allele” to match the figure legend instead of “alternate allele”. Also, in Figure 2f legend (line 757), please consider adding “, regardless in reference or individual’s allele” or equivalent at the end of the sentence to clarify the source of mechanism. I believe most readers will get confused (as I did) by conceiving the plots were showing the state of the derived alleles in Ind, not the Ref.

Reviewer #1 (Remarks to the Author):

I thank the authors to make an effort and address my previous comments. I shared the feeling of sadness that 10X Genomics decided to discontinue linked-read sequencing before its full potential is really revealed and recognized by the community.

Some additional comments below:

One suggestion that I made previously is that they should really stress the advantage of the novel computational approach. It is not just for finding variants, but for finding phased variants, given that such information is quite useful in many instances such as diagnosis of genetic diseases that are recessively inherited, and for finding mutations that interact with each other in cis (either physically by being in the same protein, or influenced by the same regulatory mechanism). Take this as an optional comment: use an example of a segment and show how two coding heterozygous variants are called in the same haplotype (you may even use IGV to show that half the linked-reads carry one allele that has two mutations in the same read). This ad hoc example may be more informative to tell users what are the main advantage of the new methods.

This idea is much appreciated. We have added a note at the end of the results section and a panel to Figure 3 that shows a nice example of a set of phased variants.

Please make sure that this is done “We will be submitting raw sequence data and assemblies to NCBI's SRA and Assembly databases”. The assembly is especially important and should not be omitted. They are unable to address the minor comments #4, but they mentioned that they have another manuscript in preparation and claimed that the public data sets do not have as high quality as the data that were generated in the current study. I think this is fine, and I hope that they can provide the data sets publicly available if the quality and quantity is much higher than previous studies.

Thank you, absolutely. Please see the Section “Data Availability” now. We had already uploaded everything to publicly accessible services in the last revision but not updated this section. This is now fixed. We will do the same thing of course for the new data in our upcoming manuscript.

Reviewer #2 (Remarks to the Author):

The authors have addressed most of my initial concerns over the revision. Pleased to see benchmark comparison against HapCUT2, as well as the MHC phasing results. Adding “reference-assisted” in the title much better communicates the identity of Aquila.

Great, we're glad we could clarify that.

Below are questions / comments on incompletely addressed concerns in the revised manuscript or response.

1. Table 5: format numbers consistently with commas. Apologies for overlooking this table in my prior review.

The reviewer has eagle eyes! We fixed the single instance of this oversight.

2. The supplementary note now states that contigs shorter than 1 kbp needs to be removed prior to evaluate the base QV. However, this means variants called from these shorter contigs be less reliable. Are these contigs also removed prior to variant calling?

Yes indeed, contigs shorter than 1 kb are removed prior to variant calling. Thank you for pointing this out, this was an oversight in our Methods. We have now added that statement to the relevant section (Variation detection for assembled contigs (Variation Module)).

3. The spectra-cn plot of the L3 assembly (NA12878) the authors attached shows good completeness overall. However, lots of false duplications (3, 4, >4x on the ~45x multiplicity), which is expected from overlapping mini contig boundaries. This is one of the common assembly errors misleading to false duplication calls if not properly handled. How are these handled? A brief discussion, along with the <1 kbp contigs from 2., should be addressed in Methods or Discussion.

Variant calls are performed only on one contig per haplotype. Contig breaks are never used in variant calling. Our minicontig boundaries abut precisely, there is never an overlapping or missing base. The “false duplications” are due to short contigs piling up at areas of assembly difficulty. These are not used for variant calling, see 2. above.

4. Although the spectra-cn plot of NA12878 seems more complete, the NA24385 L5+L6 still suffers from missing heterozygous sequences (completeness measured with Merqury was 94.2% for hap1 and 93.2% for hap2, combined hap1 + hap2 96.2%). Is this because the Chr. Y was not handled in the pipeline? Or the genome being more diverged from the reference? Are the variants on Chr. Y completely discarded, thus not creating any contigs? Stating in github page is not enough. As this is a current limitation of Aquila, it should be stated in the Results or Discussions. Likewise, if the Chr. Y was excluded for measuring Genome Fraction (%) in Table 1, clarify in the legend. Below is the spectra-cn plot and the spectra-asm plot of the combined L5+L6 contigs. The black hump at ~30x is showing the missing portion. (See attached file)

Correct, Aquila does not use the Y-chromosome, so Merqury’s statistics, which include the Y-chromosome, look worse for a male (NA24385) than a female (NA12878). We now added this statement in the Table legend: “The statistics of the female individual, NA12878, include the X chromosome; those of the male individual, NA23485, include the X chromosome but not the Y chromosome.”

5. Related to 4., the authors claim k-mer based completeness metrics are only valid when reference genomes are not available, however this underrepresents the advantage of the local de novo assembly of Aquila. Showing k-mer based metrics will complementary provide a more complete picture of the solidity that Aquila delivers as it provides an absolute measure of the genome coverage, lifting the biases caused by the reference.

Thanks for this suggestion, we have now added Supplemental Figure 3 to show the Merqury spectrum plots for the L3 assembly from female NA12878 and added this statement to the results: “k-mer spectrum analysis²⁴ indicates excellent completeness of the diploid nature of the assemblies as well (e.g., Supplementary Figure 3).”

6. I have asked on page 5, “greater than 99.5% of anybody’s two haplomes is identical to the reference” be referenced, or removed. The authors clarified in the revised text that they measure outside of the telomeres and centromeres, and responded their source of inference of the 99.5% was based on SNPs, indel, and SVs, presumably from the two human genomes used in this study genotyped with Aquila. If there is no formal reference, it has to be addressed how this number was obtained in the main text.

We addressed this in our previous response, please refer back to that. We do not infer this number from our work; it is a statement of biological fact, which is borne out by hundreds of publications on human heterozygosity. There is no single reference for it. Briefly, here is the reasoning:

The human reference sequence is mostly from an African American individual from Buffalo, New York. It was produced from BACs, which locally sampled one or the other parental haplotype of that individual. SNP heterozygosity in African Americans is about 0.2%; adding in the base-wise heterozygosity from other types of variants (especially indels of all sizes), one gets a number of about 0.5%. So, 0.5% of each comparison to the reference is different, 99.5% is the same. Of course 99.5% is a ballpark number, but it is an educated one: it’s certainly less than 100%, and no less than 99%, outside of centromeres and telomeres.

That said, to be conservative, we changed the number to 99%, which still makes the same point.

7. Page 13 line 312-314: Please use “individual’s allele” to match the figure legend instead of “alternate allele”. Also, in Figure 2f legend (line 757), please consider adding “, regardless in reference or individual’s allele” or equivalent at the end of the sentence to clarify the source of mechanism. I believe most readers will get confused (as I did) by conceiving the plots were showing the state of the derived alleles in Ind, not the Ref.

Thank you for these suggestions. We have added clarifying terms in both the text and the figure legend.

Reviewer #2 (Remarks to the Author):

The authors have fully addressed my comments and concerns. I like the optional comment from Reviewer #1, showing an example of the main advantage of Aquila, which is now present in Fig. 3c. One minor comment is to show the scale bar accordingly in that panel, or at least state how large the hap1 and hap2 contigs are spanning in the legend. Otherwise, I am happy with the revised manuscript.